# Current Biomaterials for Wound Healing

**DOI:** 10.3390/bioengineering10121378

**Published:** 2023-11-30

**Authors:** Mauricio Downer, Charlotte E. Berry, Jennifer B. Parker, Lionel Kameni, Michelle Griffin

**Affiliations:** 1Department of Surgery, Division of Plastic and Reconstructive Surgery, Stanford University School of Medicine, Stanford, CA 94305, USA; mdown18@stanford.edu (M.D.); berryc@stanford.edu (C.E.B.); jparker6@stanford.edu (J.B.P.); leokmen@stanford.edu (L.K.); 2Institute for Stem Cell Biology and Regenerative Medicine, Stanford University School of Medicine, Stanford, CA 94305, USA; 3Hagey Laboratory for Pediatric Regenerative Medicine, 257 Campus Drive, MC 5148, Stanford, CA 94305, USA

**Keywords:** wound healing, biomaterials, wound repair, skin repair

## Abstract

Wound healing is the body’s process of injury recovery. Skin healing is divided into four distinct overlapping phases: hemostasis, inflammation, proliferation, and remodeling. Cell-to-cell interactions mediated by both cytokines and chemokines are imperative for the transition between these phases. Patients can face difficulties in the healing process due to the wound being too large, decreased vascularization, infection, or additional burdens of a systemic illness. The field of tissue engineering has been investigating biomaterials as an alternative for skin regeneration. Biomaterials used for wound healing may be natural, synthetic, or a combination of both. Once a specific biomaterial is selected, it acts as a scaffold for skin regeneration. When the scaffold is applied to a wound, it allows for the upregulation of distinct molecular signaling pathways important for skin repair. Although tissue engineering has made great progress, more research is needed in order to support the use of biomaterials for wound healing for clinical translation.

## 1. Introduction

The skin is the body’s largest organ, and its accessory structures are responsible for performing several critical functions [1]. As a vital protective barrier, the skin plays a crucial role in separating the internal structures from the external environment [1]. The strategic location of the skin allows it to maintain the body’s homeostasis [1]. In order to regulate local and systemic homeostasis, the skin has developed sophisticated communication between the central nervous, endocrine, and immune systems [1]. In addition, the skin has been described as a neuro–endocrine–immune organ that calibrates adaptive mechanisms through neural and humoral pathways in response to environmental stressors or pathological processes [2].

Present knowledge has revealed that wound healing after skin injury is a complex process that involves interaction between the extracellular matrix (ECM) and multiple cell populations, including fibroblasts, keratinocytes, endothelial cells, macrophages, and platelets [3,4].(Figure 1) Equally important in wound healing is the coordinated effort of numerous growth factors (GFs), cytokines, and chemokines. Derangement in the cellular and molecular signals of wound repair can lead to healing impairment and the formation of chronic, nonhealing wounds [4]. The economic burden imposed by chronic, nonhealing wounds is substantial [5]. In the United States, chronic wounds affects around 6.5 million patients [5]. Sen et al. estimated that more than USD 25 billion is spent each year by the healthcare system on treating wound-related complications [5]. The immense impact of wounds coupled with the limited efficacy of current treatments calls for the development of novel wound therapies.

To currently restore skin defects, surgeons use autologous full-thickness or split-thickness skin grafts. When autologous split-thickness skin grafts are necessary, a thin layer of skin, including the full epidermis and part of the dermis, is shaved from a donor site [6] (Figure 2). The donor site is usually from another area of the body, such as the ventral thigh, buttocks, or upper arm [6]. Unfortunately, autologous skin grafts are limited by donor site availability. To overcome this, researchers are aiming to tissue-engineer skin replacements [7]. Biomaterials have demonstrated significant promise in enhancing skin wound-healing outcomes, with a particular emphasis on accelerating the healing process, reducing wound contracture, and scarring [7]. 

The aim of this review is to highlight important properties of biomaterials for skin repair, including current skin substitutes, and discuss new experimental skin tissue engineering techniques. While biomaterials have emerged as an area of growing scientific inquiry for skin tissue engineering in the clinical field, a comprehensive review of the pros and cons of the current biomaterials and prospects of future biomaterials is lacking. 

## 2. Properties of Biomaterials for Wound Healing

The main objective of skin tissue engineering is to produce high-quality skin replacement products to facilitate faster wound repair [8]. In the realm of skin replacement therapies, allografts and autografts stand as the primary options employed to facilitate wound healing [8]. An allograft refers to the transplantation of tissues or organs from a donor of the same species as the recipient, while an autograft is a tissue graft derived from and grafted onto the same individual. Advanced techniques using scaffolds or temporary biologic structures have increased the possible techniques to aid in advanced wound repair. Scaffolds can come in various forms, including porous, fibrous, microspheres, hydrogels, composites, and acellular structures, and are all composed of the biomaterials necessary for cell growth and differentiation analogous to the cells’ original microenvironment [8]. 

Biomaterials are used to aid in the process of wound healing by providing physical support, promoting cell growth and migration, and facilitating the formation of new tissue [8,9,10]. To effectively support these processes, biomaterials used in wound healing should possess specific mechanical properties. Biomaterials for skin regeneration include appropriate mechanical strength, flexibility, porosity, structure, biodegradability, and biocompatibility [10] (Table 1). Appropriate tensile strength helps the material withstand mechanical stress without losing shape, while flexibility allows it to conform to the wound’s shape [10]. The mechanical properties of a biomaterial should be tailored to the specific wound site, taking into account size and location [10]. 

Biomaterials used in scaffolds for skin tissue engineering can be synthetic, natural, or a composite [11]. Synthetic biomaterials can be nonbiodegradable, and these biomaterials are most commonly used for tissue engineering and include a group of aliphatic polyesters, a material that is flexible, easily processable, and nontoxic and has high mechanical strength [11]. Aliphatic polyesters are a class of polymers composed of repeating units derived from nonaromatic hydrocarbon synthetic biomaterials that encompass polymers, such as polylactic acid (PLA) and polyglycolic acid (PGA). These biomaterials have been used successfully in wound healing and as a delivery system for growth factors for aiding in skin regeneration [11,12,13]. Cacciotti et al. demonstrated the potential of polylactic acid as a delivery system. Firstly, they utilized PLA fibrous membranes (PFMs) to release bioactive molecules for tissue repair [12]. Their study assessed the antimicrobial activity and impact on the in vitro growth of human cardiac mesenchymal stem cells, both in the presence and absence of oxidative stress. They observed that these H2S-releasing PFMs could serve as cost-effective and efficient scaffolds or patches for various biomedical applications in tissue repair. Furthermore, the same team explored the use of PLA as a delivery system for treating oral mucosal diseases [13]. Cacciotti et al. assessed the cytotoxicity of 18-β-glycyrrhetic acid (GA), extracted from Glycyrrhiza glabra, on PE/CA-PJ15 cells [13]. Poly(lactic-co-glycolic acid) and chitosan were used as releasing systems for GA. Both systems were evaluated on PE/CA-PJ15 cells and normal human gingival fibroblasts (HGFs). GA-loaded nanoparticles (GA-NPs) and fibers (GA-FBs) were found to successfully release GA without any cytotoxic effects [13].

Natural biomaterials include collagen, silk, gelatin, and fibrinogen. Their resemblance to the native ECM has shown to improve skin healing [11]. Collagen is one of the most commonly used of the natural biomaterials [11]. This may be due to the vast amount of collagen found in the skin and the rest of the body. Natural biomaterials may also be of animal origin and comprised of animal derivatives, such as keratin, bovine serum albumin, and eggshell membrane [11]. Animal-derived biomaterials, such as those that incorporate keratin, can release GFs and have found to be beneficial in wound-healing applications [11]. Composite biomaterials refer to materials that are created by combining two or more different types of materials, each with distinct properties, with a matrix to achieve the specific desired characteristics for use in biomedical applications [11]. Cellulose and chitosan nanoparticles combined with cellulose poly(methylmethacrylate) fibers is an example of a composite biomaterial used in wound healing. This biomaterial has shown to be useful as an anti-infective bandage, with specific activity for *Staphylococcus aureus* [11]. 

### 2.1. Synthetic Biomaterials in Wound Healing 

Synthetic biomaterials are artificially created materials designed and engineered for use in various biomedical and health applications. Synthetic polymers have been used in different aspects of wound healing and can currently be found in suture material, coils used for aneurysms, and drug delivery and growth factor (GF) delivery systems and as scaffolds effectively increasing the rate of wound healing [14]. Wound dressings made from synthetic materials may increase wound regeneration and have been effective in different animal experiments [14]. Shimpo et al. demonstrated that the placement of PLA nanosheet material over dorsal wounds had the ability to accelerate wound healing on male mice [15]. In the study conducted by Shimpo et al., the rate of skin regeneration was compared between mice treated with a PLA nanosheet and control mice without such treatment [15]. On day 4, the experimental group exhibited the formation of a distinct, possibly collagen nanofiber-based transparent layer that was not observed in the control group [15]. This transparent layer could potentially play a crucial role in promoting wound healing. In contrast, the control group showed an increase in lymphocyte infiltration, indicating an inflammatory reaction [15]. The absence of an inflammatory response in the PLA group may be a pivotal factor contributing to the enhanced wound healing facilitated by PLA nanosheets [15]. 

Other synthetic biomaterials have exhibited success in enhancing wound regeneration in experimental animal models. PGA, another biodegradable polymer composed of repeating glycolic acid units, is one such example. In their study, Kibe et al. investigated the impact of different wound dressings on wound healing in 7-week-old rats with full-thickness defects or punch biopsies [16].

The researchers compared the efficacy of PGA dressings with collagen sponges, alongside a control group of rats that did not receive any wound dressings. To assess wound contracture in each group, they measured the levels of alpha smooth muscle actin (SMA), a marker associated with wound contraction [16]. Interestingly, the alpha SMA levels reached their peak on the fourth day of the healing process in all groups. Notably, the control group exhibited the highest levels of alpha SMA on day 4, while the PGA group displayed the lowest levels of alpha SMA on the same day [16]. Alpha SMA can be linked to increased fibrosis and scar contracture [16]. 

### 2.2. Biomaterials as a Delivery System 

As stated previously, biomaterials may aid in GF delivery systems. GFs, which are proteins, hold a pivotal role in orchestrating the intricate process of wound healing by regulating cellular activities vital for tissue repair and regeneration [17]. In the context of wound healing, these growth factors contribute to the coordination and acceleration of the overall healing procedure. Synthetic biomaterial scaffolds serve as a potential means to facilitate the delivery of growth factors to wounds, thereby enhancing tissue regeneration. A study conducted by Xie et al. demonstrated the potential of using chitosan and poly(ethylene oxide) to create a nanofiber mesh [17]. This nanofiber mesh exhibited the capability to effectively distribute growth factors, such as vascular endothelial growth factor (VEGF) and platelet-derived growth factor BB (PDGF-BB), to normal full-thickness wounds on the skin of rats. Specifically, VEGF was incorporated into the nanofiber to stimulate angiogenesis, thus increasing blood flow to support wound healing [17]. Additionally, the presence of PDGF-BB aimed to accelerate tissue remodeling and promote enhanced tissue regeneration. Preliminary findings from this research indicate that the delivery of growth factors via nanofiber scaffolds can expedite the wound-healing process by enhancing angiogenesis, facilitating re-epithelialization, and controlling the formation of granulation tissue. Furthermore, evidence suggests an earlier deposition of collagen and accelerated remodeling of the injured site [17].

### 2.3. Collagen in Wound Healing 

Collagen is a major component of the ECM and plays a significant role in the different phases of wound healing [18]. When skin is injured, the collagen exposure activates the coagulation cascade [18]. The collagen fragments mediate neutrophils to initiate an immune response and the inflammatory phase of wound healing [18]. Collagen used in biomaterials for tissue engineering is primarily derived from animal sources. Research has demonstrated that bovine collagen can improve wound healing when used as a skin substitute. Li et al. investigated the wound-healing properties of bovine bone collagen oligopeptides in an excisional wound model in mice [19]. The mice that received collagen oligopeptides had increased tensile strength and biochemical markers associated with wound healing, including albumin, prealbumin, transferrin, and hydroxyproline [19]. While collagen from animal sources has shown potential in aiding wound healing, there is a potential increased risk of infection associated with its use [20].

Collagen can be harvested from different animal sources such as rat tail tendon [18]. Rat tail tendon is known for its high concentration of type 1 collagen [18,20]. The extracted collagen can be utilized to create sponges that adhere to the wound bed and facilitate a suitable environment for skin tissue regeneration [21]. These collagen sponges are commonly employed in the treatment of ulcers, pressure sores, and burns [21]. Collagen continues to be a popular biomaterial in the field of tissue engineering due to its low immune activation, high biocompatibility, and ease of manipulation [21].

### 2.4. Cellulose in Wound Healing 

Cellulose, which is an insoluble substance responsible for the structure of a plant’s cell wall, is a natural biomaterial that could assist in tissue regeneration. As a polysaccharide made up of numerous chains of glucose, it has gained attention for its potential in wound healing because it can imitate the tissue properties required for tissue engineering [22]. Bacterial cellulose, also known as cellulose produced by bacteria, is a nanofiber material that is compatible with the human body. Its capability to retain water and maintain its mechanical strength even when wet makes it suitable for tissue engineering [23]. Cao et al. investigated the effectiveness of using bacterial cellulose in conjunction with human urine-derived stem cells for wound healing in rats [23]. The immune system of Sprague–Dawley rats was regulated through cyclosporine injections, after which 2 cm full-thickness wounds were inflicted on the dorsum. Cao et al. observed that the combination of bacterial cellulose and human urine-derived stem cells yielded better results in terms of wound healing that either material used alone [23]. Despite cellulose being a promising option for wound healing, its solubility can present challenges for clinical translation [24]. 

### 2.5. Silk in Wound Healing

Silk is a type of protein fiber produced by spiders, with variations in its composition depending on the insect species [25]. Silk is not soluble in water and is primarily made up of two types of proteins: fibroin and sericin. Fibroin protein, which is biocompatible and biodegradable, exhibits low immune activation, whereas the sericin protein has been shown to enhance tissue regeneration [25,26]. Although both silk proteins have potential for wound care, silk fibroin has been predominantly used in wound care due to its low immune activation [25,26]. Park et al. conducted a study to investigate the signaling mechanism involved in the use of silk fibroin for wound healing. Silk fibroin scaffolds were observed to upregulate transcription factor NF-KappaB (NF-kB), a transcription factor with a multifaceted role, to actively participate in all phases of the wound-healing process: inflammation, proliferation, and remodeling [27].

## 3. Cell Sources in Biomaterials for Skin Tissue Engineering 

Understanding the optimal cell source to utilize with the biomaterial for effective skin tissue engineering is as important as the biomaterial alone [8]. Stem cells, such as hair follicle stem cells (HFSCs), melanocyte stem cells (MelSCs), adipose-derived stem cells (ADSCs), fibroblasts, and keratinocytes, are all present in the skin and can be used as cell sources for skin healing [7,28,29,30,31,32,33,34,35,36,37] (Table 2). HFSCs maintain hair follicle growth, give rise to sebaceous glands, and contribute to skin reepithelization [8]. MelSCs supply melanin for hair and skin pigmentation [8]. ADSCs have shown to play a role in promoting skin repair [8]. Keratinocytes are the most abundant cells present in the epidermis and are vital for wound closure [7,8]. Having established the optimal cell type, the cells are placed into a scaffold to promote their proliferation. A scaffold is a structure, either artificial or natural, that is composed of various biomaterials and supports tissue development by providing a framework for cell attachment and growth [28]. It acts as a temporary ECM that mimics the structural and functional properties of the surrounding tissue and serves as a guide for cellular processes, such as proliferation, differentiation, and tissue remodeling. In essence, a scaffold is a critical component of tissue engineering that provides a three-dimensional environment for cells to interact with each other and their surroundings, enabling the formation of functional tissues that can be used to repair or replace damaged or diseased tissues [8,28]. The cell-seeded scaffold can then be placed over the wound to increase wound healing [8] (Figure 3). Several studies have evaluated the use of different cell types as potential cell sources for the biomaterials for skin repair. 

### 3.1. Keratinocytes 

Keratinocytes make up the outermost layer of the skin [7]. During the healing process, keratinocytes play a crucial role in epithelization and wound closure [7]. Dominguez et al. examined the effects of applying frozen sheets of human epidermal keratinocytes on murine wound healing [29]. The keratinocyte application enhanced the wound-healing process by upregulating growth factors, such as VEGF, transforming growth factor-beta, and ECM proteins [29]. 

### 3.2. Fibroblasts 

Fibroblasts are mesenchymal cells that play a significant role in wound healing. These cells secrete numerous cytokines and GFs that have an effect on skin healing in the proliferative and remodeling phases [30]. Mineo et al. performed an animal study investigating the efficacy of artificial tissue substitutes composed of hyaluronic acid, collagen, and dermal fibroblasts [31]. Their study showed that their artificial skin substitute released an increased amount of GFs, such as VEGF, increasing angiogenesis when used as an autologous graft in Sprague–Dawley Rats with deep dermal wounds [31]. Sierra-Sánchez et al. also demonstrated that dermal fibroblasts aid in the skin repair of natural skin substitutes [32].

### 3.3. ADSCs 

ADSCs are mesenchymal stem cells that have shown increasing evidence of being a key factor in wound healing. These stem cells are found in fatty depots and have the ability to release GFs that aid in skin regeneration [33]. The release of exosomes from ADSCs has also shown promise in reducing inflammation and scar tissue in wound healing [34]. Exosomes are extracellular vesicles that can be released from mesenchymal stem cells and contain proteins that can aid in the healing of chronic wounds [34]. 

ADSCs can be collected through surgical procedures, such as liposuction, from the abdomen, back, or thighs. Once collected, the ADSCs are cultured and expanded prior to being used to improve wound healing [35]. The conditioned medium from ADSCs has also been found to increase the activity of human dermal fibroblasts and keratinocytes, successfully aiding in wound healing in large animal models [35]. Ozpur et al. further demonstrated the ability to generate a skin substitute from ADSCs and keratinocytes and repair a 2 cm diameter full-thickness skin in a rat defect model [36]. 

### 3.4. HFSCs 

HFSCs have also shown great promise in the tissue engineering of skin. These cells are adult stems cells that are found in the skin [37]. One of their main functions is to produce and regenerate hair. HFSCs have the capacity for self-renewal, which entails their capability to undergo division and generate additional stem cells. Additionally, they possess the potential to differentiate into diverse cell types. This ability allows HFSCs to sustain their population and give rise to a wide range of specialized cell lineages, rendering them highly valuable for applications in regenerative medicine and tissue engineering [37]. Li et al. described their ability to differentiate into epidermal cells and sebaceous glands along with other hair cells. Additionally, HFSCs can contribute to wound healing by promoting the growth of tissue, regulating inflammation, and accelerating the formation of new blood vessels [37]. 

## 4. Skin Substitutes Currently Used in Clinical Practice

Given the encouraging role of tissue engineering strategies for wound repair, several skin substitutes are currently utilized in clinical practice [17,38,39,40,41,42,43,44,45,46]. At present, skin substitutes can be engineered using acellular materials or synthesized from various sources, such as xenogeneic or synthetic materials [6]. Each skin substitute has different advantages and disadvantages for skin repair [6] (Table 3). 

### 4.1. Acellular Skin Substitutes 

Acellular substitutes are biomaterials typically derived from human or animal tissues wherein immunogenic cellular components are removed using decellularization techniques [12]. Acellular substitutes retain their ECM, which aids in proliferation, adhesion, and differentiation [39] (Figure 4). Recently, secretomes have been investigated to increase the effectiveness of acellular skin substitutes for wound healing [40]. Secretomes are referred to as the conditioned media that stem cells are grown in. While stem cells are growing in media, they release GFs, cytokines, ECM, and other molecules that are important in wound healing. The media can then be added to biomaterials for the manufacturing of acellular skin substitutes. The addition of secretomes to acellular skin substitutes has been observed to increase vascularization [39,40]. Current commercial acellular skin substitutes used for skin repair in clinical practice include Integra and AlloDerm [6].

### 4.2. Integra and AlloDerm 

Integra is a cell-free biomaterial consisting of a collagen/glycosaminoglycan matrix with a silicone membrane. The silicone membrane acts as an epidermis, while the collagen matrix provides a suitable environment for the dermis cells to grow. Once the dermal skin has regenerated, a thin epidermal graft is placed. It was initially created for patients with severe full-thickness burn injuries [41]. Though originally developed for burn surgery, integra is now also being used for skin trauma. The silicone layer of integra allows for wound closure by preventing fluid leakage [41,42].

AlloDerm is a type of human acellular dermal matrix that is derived from cadaver human dermis preserved by freeze-drying [6]. The removal of cells from AlloDerm reduces the likelihood of an immune response and has been shown to have low rejection rates [6]. The decellularized ECM of AlloDerm retains its basement membrane, which acts as a scaffold containing collagen and elastic fibers. When applied to the wound bed, the scaffold allows the regeneration of a new dermis [6]. Rennekampff et al. investigated the effects of an acellular human dermis on wound healing in mice. Human keratinocytes were cultured onto a synthetic water-resistant dressing and applied to the full-skin defects on athymic mice. The dressing was applied with and without AlloDerm [43]. Mice in the acellular dermal matrix group showed reduced wound contraction [43]. Like Integra, AlloDerm can be used as a skin substitute for skin replacement in burn surgery [6]. 

### 4.3. Decellularized ECM

Researchers have been actively involved in the process of decellularizing natural tissues to create skin substitutes based on the ECM. These ECM-based scaffolds provide a natural microenvironment that supports cell attachment and tissue regeneration. The ECM itself holds potential as a scaffold, but challenges can arise at the wound site due to issues like oxidative stress and biofilm formation [44]. Biofilm refers to a thick layer formed by prokaryotic organisms aggregating to create a colony. Decellularizing the ECM may help address these challenges [44]. In one study conducted by Sing et al., decellularized ECM was analyzed as a potential scaffold [44]. They obtained biowaste from the submucosa of goat small intestines and then subjected it to a decellularization process. The resulting decellularized goat small intestine was transformed into a scaffold for the controlled release of curcumin, a molecule known for its ability to reduce reactive oxygen species and increase wound contractures [44]. An in vitro analysis revealed that these scaffolds possessed several advantageous properties. They were porous, biodegradable, biocompatible, antibacterial, and hydrophilic. Moreover, they demonstrated a sustained release of curcumin and exhibited the ability to scavenge free radicals [44]. 

In a study conducted by Zhang et al., the researchers showcased the application of decellularized skin/adipose flaps (DSAFs) in nude rats [45]. These DSAFs were composed of collagen and laminin, supplemented with vascular endothelial growth factors (VEGFs) and basic fibroblast growth factors (FGFs) [45]. Furthermore, they were repopulated with human umbilical vein endothelial cells and human adipose-derived stem cells [45]. When these DSAFs were transplanted onto nude rats, they underwent a process of revascularization and remodeling [45]. Notably, this process was associated with an increase in M2 macrophages, which play a crucial role in tissue remodeling and wound healing [45].

### 4.4. Cultured Epidermal Autografts

There are two kinds of autologous skin substitutes: cultured epidermal autografts (CEAs) and cultured skin substitutes (CSSs) [6]. CEAs involves culturing keratinocytes from a small sample of the patient’s own skin biopsy and transplanting the epidermal sheet to the wound area [46]. CEAs can be constructed from small regions of skin, making them a more viable option for extensive burns [47]. CEAs have been shown to be useful in a patient with a congenital nevus [46]. CEAs were used successfully to promote epithelization of the post-curettage wound [48]. Although useful, culturing keratinocytes for CEAs is still difficult, and the rate of successful take is low [6].

### 4.5. Cultured Skin Substitutes

Cultured skin substitutes (CSSs) consist of autologous keratinocytes, fibroblasts, and stem cells cultured from the patient’s own skin biopsy combined with a degradable bovine collagen matrix [46]. Living skin cells are grown in a laboratory setting before being transplanted onto the patient’s wound bed [46]. The bovine collagen matrix acts as a scaffold mimicking the ECM of skin [20,46]. As the cells grow and differentiate within the scaffold, the scaffold can help to guide the cells to create the structure needed, providing mechanical support [20,46]. The cells cultured can also secrete GFs and cytokines to stimulate tissue growth repair [7]. Similar to CEAs, CSSs can be cultured using small samples of the patient’s own skin, which is useful in cases where there is a limited donor site [46]. Compared to CEAs, a disadvantage of CSSs is their lengthier preparation time and higher cost [6].

### 4.6. Allogenic Skin Substitute

Allogenic skin biomaterials are normally produced using living neonatal foreskin fibroblasts in combination with a mesh [6,46]. This skin substitute is effective in temporarily covering deep partial and full-thickness burns, venous ulcers, diabetic foot ulcers, and chronic foot ulcers [46]. Supp et al. demonstrated that allogenic skin substitutes were effective but temporary solutions as a skin substitute [46].

The skin substitutes summarized above have all shown promise for wound healing in clinic practice. To achieve more optimal outcomes in wound healing, researchers are advancing the field by experimenting with 3D bioprinting in skin tissue engineering, experimental tissue enhancement and transplantation, and experimental wound dressings.

## 5. Experimental 3D Bioprinting Printing in Skin Tissue Engineering

The technique of bioprinting, which is a recent development in the field of tissue engineering, involves the use of a bioink solution to create 3D-printed products. The bioink solution is usually composed of a biomaterials in a hydrogel form, along with cells that are crucial for constructing a cellular microenvironment [49]. Despite the potential of 3D printing in skin tissue engineering, obstacles remain in using bioprinting for skin replacement. Specifically, the bioink materials used for the skin are often uniform in composition and may not fully replicate the complex ECM found in natural skin [49,50,51]. Kim et al. sought to investigate the use of Skin-Derived ECM (S-dECM) bioink [50]. They demonstrated the ability to successfully print a full-thickness 3D human skin model with minimal skin shrinkage [50]. Furthermore, they showed the ability to 3D-printed prevascularized skin patches that could allow for skin repair in vivo, when endothelial progenitor cells and ADSCs were added to the bioink [50]. 

An additional example of the rapid expansion of the use of 3D printing in skin tissue engineering, is the ability to heal large full-thickness wounds with a minimal amount of skin [52]. Peng et al. printed a functional unit of skin to using polyactide-co-caprolactone (PLCL) and a collagen gel. Wounds treated with the PLCL composite healed faster and exhibited the ability to regenerate skin appendages, such as hair follicles [52]. Contaminated dressings can lead to infection and necrosis and increased mortality [53]. Tsegay et al. investigated the application of 3D-printed hydrogel as an effective wound dressing [54]. Dressings were developed with a digital light processing printer. Hydrogel dressings were equipped with sensors that were capable of monitoring the pH levels and glucose concentrations of the wound. Such dressings hold great promise for patients with chronic wounds, such as diabetes [54]. 

### Experimental Tissue Enhancement in Wound Healing 

Experimental approaches for tissue enhancement have been explored to improve the success of skin substitution by providing additional support to the graft. Some examples of such approaches include the use of fetal skin cells, platelet-rich fibrin matrix (PRFM), and sheet-type reprocessed micronized acellular matrix. Poinas et al. explored the use of a collagen matrix seeded with fetal skin cells, reporting that the fetal skin cells reduced the immune rejection of the skin graft [55]. The authors concluded the cells produced GFs and cytokines that increased the rate of skin healing [55]. While this study only included a single patient, the results were promising and included reduced pain and accelerated wound healing [55]. 

PRFM, a concentrated product of platelets that naturally produces fibrin, has also shown efficacy in several areas of wound healing [56]. Reksodipurto et al. used autologous PRFM to augment skin grafts and found that it increased type 1 collagen in augmented skin grafts and improved graft take [56]. An additional method of tissue augmentation in skin substitution is the use of a sheet-type reprocessed micronized acellular dermal matrix. The skin substitute is freeze-dried and pulverized and then reprocessed into a sheet from [57]. Hahn et al. investigated the use of this novel biomaterial combined with split-thickness skin grafting in 41 patients and found that the outcomes were promising in terms of the skin loss rate (9.1%) and graft take rate (>90%) [57]. 

## 6. Wound Dressings

In addition to the experimental techniques mentioned above, wound dressings are crucial in facilitating skin regeneration and creating an environment to maintain skin repair, including silicone dressings and hydrogels. 

### 6.1. Maintenance with Molded Silicone Dressings

In regions that require skin grafts but are topographically uneven, Gotting et al. proposed the use of addition-cured silicone as a dressing for skin grafts that allowed for individualized molding [58]. The silicone was applied in a gel form that hardens quickly, with suture threads incorporated for increased stability [58]. The benefits of this proposed dressing included highly specific shaping, support and protection for the graft, and creation of a moist environment [58].

### 6.2. Wound Maintenance with Hydrogel Dressings

Hydrogel dressings are useful for wounds that require a moist healing environment and are often employed in the treatment of burns and surgical wounds and are more commonly used for chronic wounds. Their water-resistant properties make them favorable in moist environments. Interestingly, researchers are aiming to add antibiotics, antivirals, and antifungal medications to improve their efficacy for wound healing [59]. Peptide-based hydrogels have shown to be specifically beneficial due to their durability, potential to self-assemble, and capacity to replicate the ECM [60]. Yang et al. conducted a study illustrating the potential of combining hydrogel with bacterial cellulose (BC) nanofibers to stimulate cell proliferation and expedite the wound-healing process [61]. The researchers ingeniously integrated BC nanofibers into polydopamine/polyacrylamide hydrogels, thus introducing a polymer with robust BC backbones to enhance the hydrogel’s properties. These modified hydrogels exhibited noteworthy characteristics, including high tensile strength, significant tensile strain capacity, and optimal compressive properties [61].

As a result of these enhanced attributes, the hydrogel dressing demonstrated reduced inflammatory responses and accelerated wound healing in vivo when tested on wound-healing rats [61]. Yang et al. ultimately concluded that hydrogels are versatile in function, offering stable coverage, minimal displacement, and a swifter wound-healing process [61].

## 7. Challenges in Skin Tissue Engineering 

Although tissue engineering for skin has made many advances, there are still a number of challenges that are present today [62,63,64,65,66]. Limitations arise with vascularization, systemic diseases, and immune rejection, and there is difficulty in regenerating skin appendages [62,67,68,69,70]. Without an effective vasculature, the cells that are implanted in the biomaterial are subjected to tissue hypoxia [69]. The poor vasculature of skin biomaterials presents a major challenge in tissue engineering [69]. The immune system remains a major limitation in skin tissue engineering as the immune system can negatively affect skin tissue engineering through the upregulation of a foreign body response. Engineered skin constructs have the potential to elicit immune responses, which can result in scaffold rejection or inflammatory reactions. Immunological tolerance or effective immune evasion strategies remain a challenging aspect in this context [71]. The complexity of the skin structure remains a significant challenge in the field of skin tissue engineering. Certain structures, particularly skin appendages, pose difficulties in replication when utilizing biomaterials for this purpose [72].

### 7.1. Limitation in Vascularization in Skin Tissue Engineering 

Vascularization is important for adequate wound healing [62]. GFs, such as VEGF and FGF, increase vascularization and have therefore been applied to skin biomaterials [62]. Amirsadeghi et al. aimed to release these GFs at a steady state from a skin substitute to improve vascularization [62]. Although GFs may be a solution to a lack of vascularization in tissue engineering, the speed of angiogenesis can also delay wound healing [62]. To overcome this, VEGF and FGF are being added to the biomaterials to stimulate the production of new blood vessels by inducing endothelial cells to migrate to the site of the wound bed [62]. Systemic illnesses also plays a role in the lack of vascularization necessary for skin tissue engineering [62]. 

### 7.2. Immune Rejection in Skin Tissue Enginneering

The immune response to biomaterials remains a major issue when considering skin replacements. Immune cells, including macrophages, can cause an extensive inflammatory response leading to improper healing when using grafts or scaffolds [73,74]. Scientists are searching for ways to decrease the immune inflammatory response in tissue engineering. Macrophages have been classified into two types [71]: M1 macrophages are considered to be inflammatory, and M2 macrophages are anti-inflammatory. Previous research has shown that M2 macrophages aid in tissue regeneration. 

### 7.3. Limitations in Skin Appendage Regeneration 

Tissue engineering using biomaterials can improve wound healing, but it is still difficult to regenerate skin appendages. Skin appendages include nails, hair follicles, and glands [72]. The regeneration of sebaceous glands and hair follicles is an active area of research when considering skin biomaterials. Zhang et al. aimed to improve skin appendages by evaluating adult scalp dermal progenitor cells and epidermal stem cells with a type 1 collagen scaffold in full-thickness wounds in adult nude mice [75]. The study demonstrated that the regeneration of skin appendages could be possible but it is highly dependent on the timing of the stem cell culture [75]. 

## 8. Conclusions

Skin tissue engineering using biomaterials has become an important technique for treating wounds and burns, and the field is continuously evolving [68]. Researchers are investigating ways to decrease contraction and promote wound healing [68]. Utilizing PGA nanofibers resulted in a reduction in alpha SMA expression, subsequently reducing wound contracture in dorsal wounds of mice [16]. Ongoing research endeavors aim to explore methods for enhancing wound closure in comparison to the control group. Optimizing the cellular delivery vehicle to induce more re-epithelization is an active area of focus [76]. The application of biomaterials for delivering GFs and drugs in wound healing remains a valuable approach for promoting wound regeneration [17]. Researchers are actively exploring nanoscale drug delivery systems and innovative biomaterials that could offer improved compatibility with drug delivery techniques [17].

Biomaterials are continually being developed and modified to create scaffolds that are more effective in promoting cell growth and tissue regeneration. Novel materials, such as 3D-printed hydrogels, are being actively explored for their use in skin tissue engineering [11,50,77]. By focusing on the unique requirements of skin tissues, researchers can create more effective and efficient strategies [11,78]. Researchers are exploring ways to harness the regenerative properties of stem cells and use them to repair or replace damaged or diseased tissues [79]. Microfluidic technologies are emerging as a promising technique for generating dynamic and precise microenvironments for cell growth and tissue regeneration [80]. 

Chronic wounds are a significant economic burden on patients in the United States [81]. Using biomaterials for skin tissue engineering holds great potential to decrease the burden of chronic wounds. However, greater research is needed to fully commit to tissue engineering as a potential clinical strategy to combat nonhealing wounds. With exciting ongoing work to continue improving applications of biomaterials in wound healing, tissue engineering holds promise as a key strategy for improving wound healing.

## Figures and Tables

**Figure 1 bioengineering-10-01378-f001:**
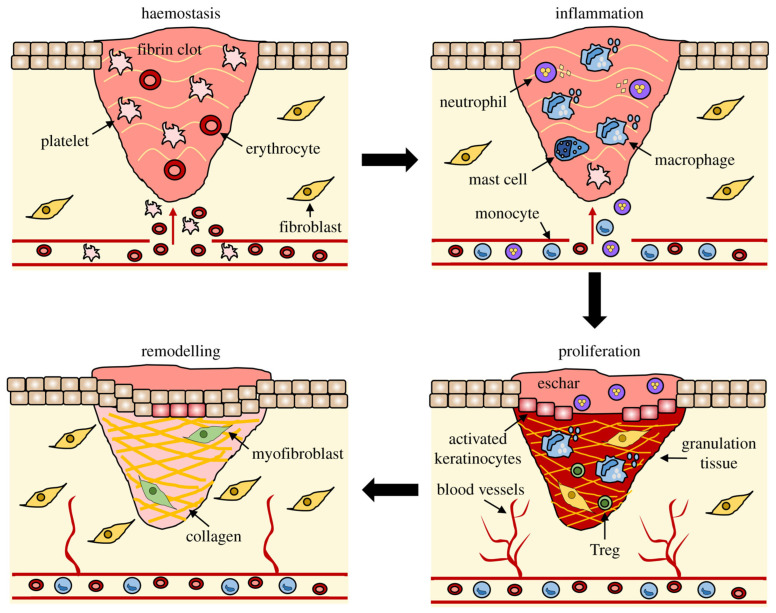
The stages of wound healing. The stages of wound healing involve an interaction between the extracellular matrix and multiple cell populations, including platelets, fibroblasts, keratinocytes, endothelial cells, and macrophages. Taken with permission from [4].

**Figure 2 bioengineering-10-01378-f002:**
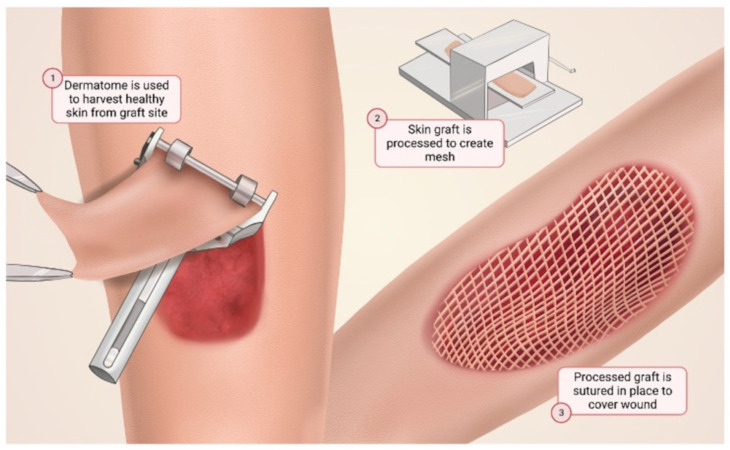
The use of autografts in wound healing. An autograft may be necessary for burn victims or patients with large wounds. A sheet of skin is harvested from another area of the body on the patient. The harvested skin is processed to create a mesh and placed over the burn or large wound, thus promoting increased wound healing from the keratinocytes within the harvested autograft.

**Figure 3 bioengineering-10-01378-f003:**
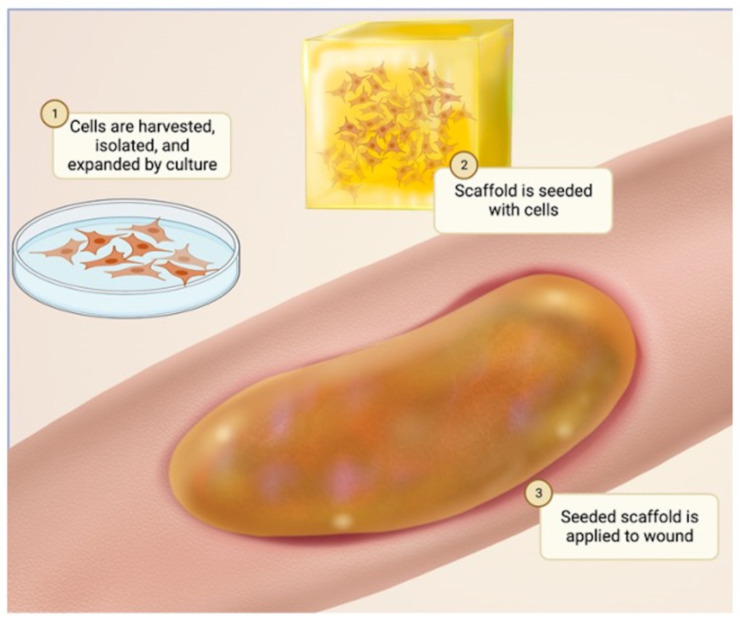
The use of cells in tissue engineering for wound healing. Stem cell seeding, also known as stem cell transplantation from the skin of a donor, involves organizing stem cells into a scaffold and giving them time to proliferate. Once proliferation is complete, the seeded scaffold is placed over the wound. The rapid proliferation of stem cells can then aid in wound healing.

**Figure 4 bioengineering-10-01378-f004:**
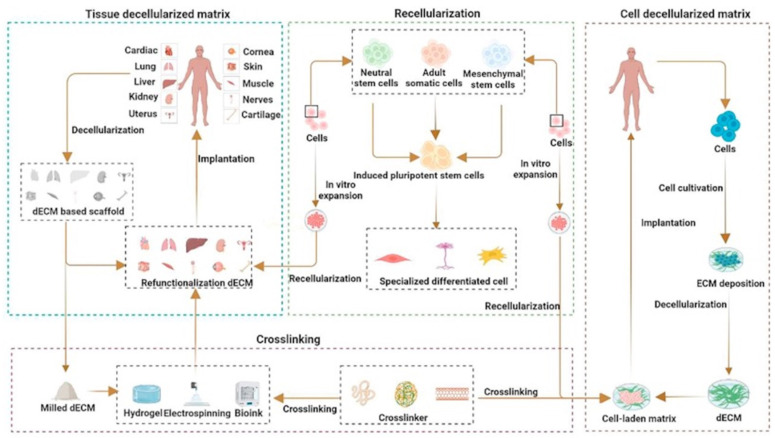
Decellularized technique to make acellular tissue graft. Functional decellularized extracellular matrix used for tissue repair. Taken with permission from [39].

**Table 1 bioengineering-10-01378-t001:** Properties of biomaterials in wound healing.

Property	Effect on Wound Healing
Biocompatibility	Biomaterials need the ability to interact with biological systems without causing harm Should not trigger adverse reactions within biological systems Their use is characterized by the absence of harm or negative impact on biological entities [8,10]
2. Biodegradable	Biomaterials should break down slowly over time, matched to the regeneration of skin following injury Should breakdown into harmless components Biodegradability facilitates the natural breakdown of the material into nonharmful elements [8,10]
3. Porosity	Pores facilitate cell infiltration Allows for nutrient and oxygenation diffusion Allows for waste removal [8,10]
4. Mechanical strength	Influence on mechanical forces and structural integrity Fracture resistance Load-bearing capacity Should have similar mechanical properties to the tissue being replaced [8,10]
5. Scaffold structure	Important to determine cell growth Important to determine tissue regeneration Key considerations include: Architecture Interconnectivity Hydrophobicity Hydrophilicity [8,10]

**Table 2 bioengineering-10-01378-t002:** Cell sources for skin tissue engineering.

Cell Type	Key Properties
Keratinocytes	Crucial roles in the process of skin regeneration Important for integration with other cell types Development of more intricate skin equivalents Important for epithelization of skin
2. Fibroblasts	Essential in producing vital components of the extracellular matrix including collagen [8,30]
3. Mesenchymal cells	Crucial role in supporting tissue repair Contribute to the formation of functional skin structures Assist in immune responses Reduce inflammation Ability to differentiate into various cell lineages [7,8]
4. Hair follicle stem cells	Particularly effective in facilitating epidermal regeneration Facilitate interactions with other cells needed for effective wound healing [8,37]

**Table 3 bioengineering-10-01378-t003:** Skin substitutes used in clinical practice.

Skin Substitute	Key Properties	Examples
Allograft	Utilizes skin tissue from a different individual of the same species Sourced from a human donor, typically cadaveric Ensures a readily available supply of healthy and viable skin for transplantation Employed for wounds, burns, or injuries where autografts are impractical Particularly beneficial for cases requiring temporary coverage Useful when extensive autograft donor sites are limited [41,42]	Cadaveric Allograft Skin tissue obtained from deceased human donors [41] Homograft Another term for allograft, emphasizing the use of tissue from the same species [41] AlloDerm A brand name for human acellular dermal matrix allograft [42] AlloSkin Commercial allograft product used for skin grafting [42]
Cultured Epithelial Autograft (CEA)	Advanced skin substitute crafted through in vitro cultivation Utilizes the patient’s own epithelial cells Consists of cultured epithelial cells for skin repair Offers a cell-based alternative in the field of skin regeneration Allows amplification of a small biopsy into a more extensive skin graft Overcomes the constraint of limited donor sites effectively [46]	Cultivated from the patient’s epidermal cells Used for skin repair and regeneration Composite CEA Combination of cultured epithelial cells and other cell types Aimed at enhancing the complexity and functionality of the graft Cryopreserved CEA Cultured epithelial autografts preserved through cryopreservation Allows for long-term storage and flexibility in clinical use [46]
Synthetic Skin Substitutes	Replicate the structure and functionality of natural skin Specifically designed for applications in skin tissue engineering Comprise various synthetic materials Biomimetic scaffolds Bioactive components [16,17]	Biodegradable Synthetic Matrices: Composed of materials including polyglycolic acid (PGA) and polylactic acid (PLA), these break down over time and serve as temporary coverings for wounds Polymeric Films: Thin films made of synthetic polymers, such as polyurethane or silicone, create a protective barrier for wounds and promote a moist environment conducive to healing Hydrogels: Water-absorbing polymers that create a moist environment for wound healing, often containing synthetic components like polyethylene oxide or polyvinyl alcohol Artificial Dermal Substitutes: Designed to replicate the dermal layer of skin, these substitutes often consist of synthetic materials, like nylon or silicone, and are used alongside other wound-care techniques Bioengineered Skin: Combining synthetic and biological components, these substitutes involve a synthetic matrix populated with living cells, aiming to closely mimic the structure and function of natural skin Synthetic Nanofiber Scaffolds: Nanofiber scaffolds made from synthetic materials, like poly(ε-caprolactone) (PCL), provide a three-dimensional structure for cell attachment and growth, contributing to wound healing [16,17]

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
