# Peer review of "Current Biomaterials for Wound Healing"

_bioengineering, 2023, doi:10.3390/bioengineering10121378_

Round 1
Reviewer 1 Report (New Reviewer)
Comments and Suggestions for Authors
General comments
The submitted manuscript consists in a review paper about the biomaterials currently available for wound healing applications.
Tables have to be prepared and added in order to provide a summary about the main outputs to the readers, for the most important paragraphs. On the other hand, Table 1 could be removed, since it ia not so significant and it is better to report the same information within the body text.
Finally an accurate English grammar revision is strongly recommended.
Detailed comments and remarks are listed below point by point.
1. Introduction
- At the end of the Introduction section the originality and added value to the scientific community of the present review paper has to be evidenced.
-
2. Properties of biomaterials for wound healing
The following sentences “Aliphatic polyesters are a class of polymers composed of repeating units derived from non-aromatic hydrocarbons osynthetic biomaterials that encompass polymers such as Polylactic acid (PLA), Polyglycolic acid (PGA) and others have been used successfully in wound healing, and as a delivery system for growth factors to aid in skin regeneration.11” have to be supported with further references, including “Hydrogen sulfide-releasing fibrous membranes: Potential patches for stimulating human stem cells proliferation and viability under oxidative stress. International Journal of Molecular Sciences, 19(8) (2018), 2368.” and “Controlled release of 18-β-glycyrrhetic acid by nanodelivery systems increases cytotoxicity on oral carcinoma cell line. Nanotechnology, 29(28) (2018), 285101.”
- The reported definition and concept of composite (“Composite biomaterials can be composed of different combinations of natural and synthetic biomaterials”) is not so correct….usually composites are composed of a matrix and a filler, combining materials from different categories, not one natural and the other one synthetic. Please better explain this point.
7.0. Challenges in Skin tissue engineering
Table 1 has no sense. It can be removed reporting the considerations within the text body.
8.0 and 8.1
It is suggested to combine Paragraphs 8.0 and 8.1, since both of them are very short.
Comments on the Quality of English Language
The English quality is good
Author Response
Date: November 6th, 2023
Re: Update in the Use of Biomaterials for Wound Healing
Manuscript Number: bioengineering-2358887
To: The Editors at Bioengineering:
We thank the reviewers for their careful consideration of our manuscript. Their thoughtful and constructive comments have led to a significantly revised and improved manuscript. Each reviewer’s comments are addressed below, including references to the corresponding changes to the manuscript.
Reviewer comments are provided below in bold black font. Our responses are provided in blue font. New text is in red font.
Thank you for the consideration of this manuscript.
Sincerely,
Dr Michelle Griffin MBChB PhD
Reviewer 1
- Introduction
- At the end of the Introduction section the originality and added value to the scientific community of the present review paper has to be evidenced.
Thank you for your comment. We agree with the reviewer and have made the necessary changes suggested and can be seen below and on page: 2 .
The aim of this review is to highlight important properties of biomaterials for skin repair, current skin substitute methods, and new experimental skin tissue engineering techniques, that may be useful in increasing techniques necessary for proper wound care.
- Properties of biomaterials for wound healing
The following sentences “Aliphatic polyesters are a class of polymers composed of repeating units derived from non-aromatic hydrocarbons osynthetic biomaterials that encompass polymers such as Polylactic acid (PLA), Polyglycolic acid (PGA) and others have been used successfully in wound healing, and as a delivery system for growth factors to aid in skin regeneration.11” have to be supported with further references, including “Hydrogen sulfide-releasing fibrous membranes: Potential patches for stimulating human stem cells proliferation and viability under oxidative stress. International Journal of Molecular Sciences, 19(8) (2018), 2368.” and “Controlled release of 18-β-glycyrrhetic acid by nanodelivery systems increases cytotoxicity on oral carcinoma cell line. Nanotechnology, 29(28) (2018), 285101.”
- The reported definition and concept of composite (“Composite biomaterials can be composed of different combinations of natural and synthetic biomaterials”) is not so correct….usually composites are composed of a matrix and a filler, combining materials from different categories, not one natural and the other one synthetic. Please better explain this point.
Thank you for your comment. We agree with the reviewer and have made the necessary changes suggested and can be seen below and on page: 4 .
Composite biomaterials refer to materials that are created by combining two or more different types of materials, each with distinct properties, with a matrix to achieve specific desired characteristics for use in biomedical applications.11 Cellulose chitosan nanoparticles combined with cellulose poly(methylmethacrylate) fibers is an example of a composite biomaterial used in wound healing. This biomaterial has shown to be useful as an anti-infective bandaged, with specific activity for Staphylococcus aureus.11
- 7.0 Challenges in Skin tissue engineering
Table 1 has no sense. It can be removed reporting the considerations within the text body.
Thank you for your comment. We agree with the reviewer and have deleted table 1.
- 8.0 and 8.1
It is suggested to combine Paragraphs 8.0 and 8.1, since both of them are very short.
Thank you for your comment. We agree with the reviewer and have combined both sections.
Reviewer 2 Report (Previous Reviewer 1)
Comments and Suggestions for Authors
Based on the thorough and satisfactory revisions made to the manuscript in response to the reviewers' comments, I recommend accepting it for publication. The author has successfully addressed all concerns raised by the reviewers, making the manuscript worthy of publication in its current form.
Author Response
- Based on the thorough and satisfactory revisions made to the manuscript in response to the reviewers' comments, I recommend accepting it for publication. The author has successfully addressed all concerns raised by the reviewers, making the manuscript worthy of publication in its current form.
Thank you for your comment, we appreciate your recommendation on acceptance of this article.
Round 2
Reviewer 1 Report (New Reviewer)
Comments and Suggestions for Authors
General comments
The Authors partially applied the Referee’s suggestions and remarks.
- They did not prepare and add Tables necessary to provide a summary about the main outputs to the readers, for the most important paragraphs. They are very important in a review paper.
- The Authors did not properly reply to the following request “At the end of the Introduction section the originality and added value to the scientific community of the present review paper has to be evidenced”. They reported the aim and not the originality with respect other review papers about the same topic.
- They did not consider the following suggestion “The following sentences “Aliphatic polyesters are a class of polymers composed of repeating units derived from non-aromatic hydrocarbons osynthetic biomaterials that encompass polymers such as Polylactic acid (PLA), Polyglycolic acid (PGA) and others have been used successfully in wound healing, and as a delivery system for growth factors to aid in skin regeneration.11” have to be supported with further references”.
Comments on the Quality of English Language
The English language quality is good.
Author Response
Date: November 19th, 2023
Re: Update in the Use of Biomaterials for Wound Healing
Manuscript Number: bioengineering-2358887
To: The Editors at Bioengineering:
We thank the reviewers for their careful consideration of our manuscript. Their thoughtful and constructive comments have led to a significantly revised and improved manuscript. Each reviewer’s comments are addressed below, including references to the corresponding changes to the manuscript.
Reviewer comments are provided below in bold black font. Our responses are provided in blue font. New text is in red font.
Thank you for the consideration of this manuscript.
Sincerely,
Dr Michelle Griffin MBChb PhD
Reviewer 1
1.They did not prepare and add Tables necessary to provide a summary about the main outputs to the readers, for the most important paragraphs. They are very important in a review paper.
Thank you for your comment. We agree with the reviewer and have made the necessary changes suggested and can be seen below and on page: 17.
Table 1: Properties of Biomaterials in Wound Healing
|
Property |
Effect on wound healing |
|
1. Biocompatibility |
· Biomaterials need the ability to interact with biological systems without causing harm · Should not trigger adverse reactions within biological systems · Their use is characterized by the absence of harm or negative impact on biological entities8,10 |
|
2. Biodegradable |
· Biomaterials should break down slowly over time, matched to the regeneration of skin following injury · Should breakdown into harmless components · Biodegradability facilitates the natural breakdown of the material into non-harmful elements 8,10 |
|
3. Porosity |
· Pores facilitate cell infiltration · Allows for nutrient and oxygenation diffusion · Allows for Waste removal 8,10 |
|
4. Mechanical strength |
· Influence on Mechanical Forces and Structural Integrity · Fracture resistance · Load-bearing capacity · Should have similar mechanical properties to the tissue being used to replace8,10 |
|
5. Scaffold structure |
· Important to determine cell growth · Important to determine tissue regeneration · Key considerations include: a) Architecture b) Interconnectivity c) Hydrophobicity d) Hydrophilicity8,10
|
Table 2: Cell Sources For Skin Tissue Engineering
|
Cell type |
Key Properties |
|
1. Keratinocytes |
· Crucial roles in the process of skin regeneration. · Important for integration with other cell types · Development of more intricate skin equivalents · Important for epithelization of skin |
|
2. Fibroblast |
· Essential in producing vital components of the extracellular matrix including collagen8,30 |
|
3. Mesenchymal cells |
· Crucial role in supporting tissue repair · Contribute to the formation of functional skin structures · Assist in immune responses · Reduce inflammation · Ability to differentiate into various cell lineages7,8 |
|
4. Hair follicle stem cells |
· Particularly effective in facilitating epidermal regeneration · Facilitate interactions with other cells needed for effective wound healing8,37 |
Table 3: Skin Substitutes Used in Clinical Practice
|
Skin Substitute |
Key Properties |
Examples |
|
Allograft |
· Utilizes skin tissue from a different individual of the same species · Sourced from a human donor typically cadaveric · Ensures a readily available supply of healthy and viable skin for transplantation · Employed for wounds, burns, or injuries where autografts are impractical. · Particularly beneficial for cases requiring temporary coverage · Useful when extensive autograft donor sites are limited41,42 |
· Cadaveric Allograft - Skin tissue obtained from deceased human donors41 · Homograft - Another term for allograft, emphasizing the use of tissue from the same species41 · AlloDerm - A brand name for human acellular dermal matrix allograft42 · AlloSkin -Commercial allograft product used for skin grafting42
|
|
Cultured Epithelial Autograft (CEA) |
· Advanced skin substitute crafted through in vitro cultivation · Utilizes the patient's own epithelial cells · Consists of cultured epithelial cells for skin repair · Offers a cell-based alternative in the field of skin regeneration · Allows amplification of a small biopsy into a more extensive skin graft · Overcomes the constraint of limited donor sites effectively46 |
· Cultivated from the patient's epidermal cells. - Used for skin repair and regeneration · Composite CEA - Combination of cultured epithelial cells and other cell types - Aimed at enhancing the complexity and functionality of the graft · Cryopreserved CEA - Cultured epithelial autografts preserved through cryopreservation. - Allows for long-term storage and flexibility in clinical use46 |
|
Synthetic Skin substitutes |
· Replicate the structure and functionality of natural skin · Specifically designed for applications in skin tissue engineering · Comprise various synthetic materials · Biomimetic scaffolds · Bioactive components16,17 |
· Biodegradable Synthetic Matrices: Composed of materials including polyglycolic acid (PGA) and polylactic acid (PLA), these break down over time and serve as temporary coverings for wounds · Polymeric Films: Thin films made of synthetic polymers such as polyurethane or silicone create a protective barrier for wounds and promote a moist environment conducive to healing · Hydrogels: Water-absorbing polymers that create a moist environment for wound healing, often containing synthetic components like polyethylene oxide or polyvinyl alcohol.
· Artificial Dermal Substitutes: Designed to replicate the dermal layer of skin, these substitutes often consist of synthetic materials like nylon or silicone and are used alongside other wound care techniques · Bioengineered Skin: Combining synthetic and biological components, these substitutes involve a synthetic matrix populated with living cells, aiming to closely mimic the structure and function of natural skin · Synthetic Nanofiber Scaffolds: Nanofiber scaffolds made from synthetic materials like poly(ε-caprolactone) (PCL) provide a three-dimensional structure for cell attachment and growth, contributing to wound healing16,17 |
2.0 The Authors did not properly reply to the following request “At the end of the Introduction section the originality and added value to the scientific community of the present review paper has to be evidenced”. They reported the aim and not the originality with respect other review papers about the same topic.
Thank you for your comment. We agree with the reviewer and have made the necessary changes suggested and can be seen below and on page: 4 .
The aim of this review is to highlight important properties of biomaterials for skin repair, including current skin substitutes, and discuss new experimental skin tissue engineering techniques. While biomaterials have emerged as an area of growing scientific inquiry for skin tissue engineering in the clinical field, a comprehensive review of the pros and cons of current biomaterials and prospects of future biomaterials is lacking.
3.0 They did not consider the following suggestion “The following sentences “Aliphatic polyesters are a class of polymers composed of repeating units derived from non-aromatic hydrocarbons osynthetic biomaterials that encompass polymers such as Polylactic acid (PLA), Polyglycolic acid (PGA) and others have been used successfully in wound healing, and as a delivery system for growth factors to aid in skin regeneration.11” have to be supported with further references”.
Thank you for your comment. We agree with the reviewer and have made the necessary changes suggested and can be seen below and on page: 5.
Aliphatic polyesters are a class of polymers composed of repeating units derived from non-aromatic hydrocarbon synthetic biomaterials that encompass polymers such as Polylactic acid (PLA) and Polyglycolic acid (PGA). These biomaterials have been used successfully in wound healing, and as a delivery system to for growth factors to aid in skin regeneration.11-13 Cacciotti et al. demonstrated the potential of polylactic acid as a delivery system. Firstly they utilized PLA fibrous membranes (PFM) to release bioactive molecules for tissue repair.12 Their study assessed the antimicrobial activity and impact on in vitro growth of human cardiac mesenchymal stem cells, both in the presence and absence of oxidative stress. They observed that these H2S-releasing PFMs could serve as cost-effective and efficient scaffolds or patches for various biomedical applications in tissue repair. Furthermore, they further explored the use of PLA as a delivery system for treating oral mucosal diseases.13 Cacciotti et al assessed the cytotoxicity of 18-β-glycyrrhetic acid (GA), extracted from Glycirrhiza glabra, on PE/CA-PJ15 cells.13 Poly(lactic-co-glycolic acid) and chitosan were used as releasing systems for GA. Both systems were evaluated on PE/CA-PJ15 cells and normal human gingival fibroblasts (HGFs). GA-loaded nanoparticles (GA-NPs) and fibers (GA-FBs) were found to successfully release GA without any cytotoxic effects.13
Round 3
Reviewer 1 Report (New Reviewer)
Comments and Suggestions for Authors
The Authors have strongly improved their review paper and it can be accepted in the current version.
Comments on the Quality of English LanguageThe English language is good
This manuscript is a resubmission of an earlier submission. The following is a list of the peer review reports and author responses from that submission.
Round 1
Reviewer 1 Report
Comments and Suggestions for Authors
General Comment: This paper provides a comprehensive review of the use of biomaterials for wound healing. The overall content of the manuscript seems to focus heavily on commercial skin grafts. It would be beneficial to include a paragraph discussing natural biomaterials such as collagen or decellularized extracellular matrix (dECM), as well as synthetic biomaterials. Furthermore, as the overall content of the review mainly introduces different types of skin grafts and methods of transplantation, it may be necessary to modify the title of the paper if the authors want to emphasize these aspects.
Specific Comments:
1) It appears that an overall verification of paragraph numbering is necessary.
2) Paragraph 3 requires additional review of other cell sources that constitute the skin, such as fibroblasts or adipose-derived stem cells, in addition to keratinocytes.
3) In paragraph 6, the author mentions certain limitations, but the reviewer believes that there are many other limitations such as immune rejection of skin grafts or lack of consideration for skin appendages during skin regeneration. Therefore, additional discussion and analysis of these limitations, as well as potential future directions for addressing them, should be supplemented.
4) In relation to the development of biomaterials for skin regeneration, the reviewer recommends referencing the following articles:
- Kim, Byoung Soo, et al. "3D cell printing of in vitro stabilized skin model and in vivo pre-vascularized skin patch using tissue-specific extracellular matrix bioink: A step towards advanced skin tissue engineering." Biomaterials 168 (2018): 38-53.
Author Response
Date: 5/7/2023
Re: Update in the Use of Biomaterials for Wound Healing
Manuscript Number: bioengineering-2358887
To: The Editors at Bioengineering:
We thank the reviewers for their careful consideration of our manuscript. Their thoughtful and constructive comments have led to a significantly revised and improved manuscript. Each reviewer’s comments are addressed below, including references to the corresponding changes to the manuscript.
Reviewer comments are provided below in bold black font. Our responses are provided in blue font. New text is in red font.
Thank you for the consideration of this manuscript.
Sincerely,
Dr Michelle Griffin MBChb PhD
Reviewer 1
1.It appears that an overall verification of paragraph numbering is necessary.
Thank you for your review. We have verified the paragraph numbering.
2.Paragraph 3 requires additional review of other cell sources that constitute the skin, such as fibroblasts or adipose-derived stem cells, in addition to keratinocytes.
Thank you for that feedback, we have incorporated further sections in the context of wound healing, specifically the contributions of adipose derived stem cells and fibroblast within tissue regeneration as can be seen below on pages:9-11.
3.1 Keratinocytes in wound healing
The structure of keratinocytes typically determines the various layers that make up outermost layer of the skin.6 These cells have been extensively studied for their role in wound healing and have been investigated in combination with biomaterials. Cells located in the lower skin layer divide to produce keratinocytes, which then shift to the upper layer of skin and become dead cells on the surface of the skin.6 During the healing process keratinocytes play a crucial role in the proliferative stage. They travel to the damaged dermis and release GFs such as VEGF, transforming growth factor beta, and fibroblast growth factor. These GFs promote angiogenesis and the production of connective tissue matrix facilitating tissue regeneration.6,22 Collecting keratinocytes and placing them into a scaffold helps to promote their proliferation. A scaffold is a structure, either artificial or natural, that is composed of various biomaterials and supports tissue development by providing a framework for cell attachment and growth.23 It acts as a temporary ECM that mimic structural and functional properties of the surrounding tissue and serves as a guide for cellular processes such as proliferation, differentiation, and tissue remodeling. In essence, a scaffold is a critical component of tissue engineering that provides a three dimensional environment for cells to interact with each other and their surroundings, enabling the formation of functional tissues that can be used to repaired or replaces damaged or diseased tissues.7,23 The cell seeded scaffold can then be placed over the wound to increase wound healing.7 (Figure 2)
- Fibroblast In wound healing
Fibroblasts are mesenchymal cells that may play a significant role in wound healing. These cells secrete numerous cytokines and GFs that may have an effect on skin proliferation and formation.24 There are four phases of wound healing, and fibroblast may be involved in majority of steps. The phases are hemostasis, inflammation, proliferation, and remodeling. desJardins-Park et al. stated that fibroblast may play a role in the deposition of ECM within the last three phases respectively.25 Fibroblasts aren’t a homogenous structure but in fact heterogenous and consist of different subpopulations based on their phenotype and functions.26 Regarding tissue engineering, the incorporation of dermal fibroblast, a specific type of fibroblast, into biomaterials has demonstrated encouraging outcomes for the tissue regeneration procedure. Mineo et al. performed an animal study investigating the efficacy of artificial tissue substitute composed of hyaluronic acid, collagen, and dermal fibroblast.27 Their study showed that their artificial skin substitute released an increased amount of GFs such as VEGF, therefore increasing angiogenesis when used as an autologous graph in Sprague Dawley Rats with deep dermal wounds.27 Sierra-sanchez et al. highlighted that dermal fibroblast may aid in the growth of more natural skin substitutes.28
3.3 ADSCs in wound healing
ADSCs are mesenchymal stem cells that have shown increasing evidence of being a key factor in wound healing. These pluripotent stem cells are found in fatty depots and may have the ability to release GFs that aid in tissue regeneration.29 The release of exosomes from ADSC has shown promise in reducing inflammation and scar tissue in wound healing.30 Exosomes are extracellular vesicles that may be released from mesenchymal stem cells, and contain proteins that may aid in the healing of chronic wounds.30 ADSCs may be collected through surgical procedures such liposuction from the abdomen, back or thighs. Once collected the ADSCs are cultured and may be used to aid in wound healing due to their plasticity.31 The ADSCs may be conditioned into media and have shown to increase activity of human dermal fibroblast and keratinocytes, successfully aiding in wound healing in large animal models.31 Ozpur et al. were able to generate a skin substitute from ADSCs and Keratinocytes, which increased wound healing in Wistar albino rats with a 2 cm diameter full thickness skin defect.32
3.4 HFSCs on wound healing
HFSCs have shown promise in tissue engineering. These cells are adult stems cells that may be found in the skin.33 One of their main functions is to produce and regenerate hair. HFSCs have the capacity for self-renewal, which entails their capability to undergo division and generate additional stem cells. Additionally, they possess the potential to differentiate into diverse cell types apart from their stem cell state. This ability allows HFSCs to sustain their population and give rise to a wide range of specialized cell lineages, rendering them highly valuable for applications in regenitve medicine and tissue engineering.33 Li et al. described their ability to differentiate into epidermal cells and sebaceous glands along with other hair cells. Additionally HFSCs may contribute to wound healing by promoting growth of tissue, regulating inflammation and accelerating formation of new blood vessels.33 This is done through the release of cytokines from HFSCs due to mechanical stretching done on the wound bed. Chu et al. found that mechanical stretch on the skin of mice caused the release of certain cytokines that activated M2 macrophages that lead to the release of GFs that lead to hair growth in wounds.34 With the regeneration of the hair follicle, there may be an association with the turnover of the whole skin.33
- In paragraph 6, the author mentions certain limitations, but the reviewer believes that there are many other limitations such as immune rejection of skin grafts or lack of consideration for skin appendages during skin regeneration. Therefore, additional discussion and analysis of these limitations, as well as potential future directions for addressing them, should be supplemented.
Thank you for your review and feedback, we have included additional sections on the limitations of biomaterials in wound healing. As can be seen below on pages:25-26.
6.3 Immune rejection
Immune response remains a major issue when using biomaterials in tissue engineering. The process of transferring biomaterials on to a wound can cause an increase in activity of the patient’s immune system. An important action of the immune system is the removal of cellular debris, and this may happen at the sight of the wound. Immune cells such as macrophages may cause extensive tissue damage and improper healing when using grafts or scaffolds.76,77 Scientist are searching for ways to decrease the immune inflammatory response in tissue engineering. Ma et al. investigated the use of a titanium plate in dental implants to regulate the immune activity of macrophages within the implant. Macrophages have a dual purpose in wound healing.78 M1 macrophages are inflammatory and M2 macrophages are anti-inflammatory, previous research has shown that M2 macrophages aid in tissue regeneration. The titanium implant showed an increase in M2 macrophages in dental implants. This may be a useful technique in regulating immune rejection in the tissue engineering.77,78
6.4 Limitations in Skin appendage regeneration
Tissue engineering may be able to increase regeneration in wound healing, but complications still arise when it comes to regeneration of skin appendages. Skin appendages include nails, hair follicles and glands. Each skin appendage has a specific function and without it, skin may not be able to function the way it normally does. This may lead to decrease in the protection the skin usually provides.79 Regeneration of sebaceous glands and hair follicles is an area of research that may be the next step for tissue engineering. Zhang et al. sought to use tissue engineered skin substitutes from adult scalp dermal progenitor cells and epidermal stem cells along with a type 1 collagen scaffold on adult nude mice with full thickness wounds, in hope to increase wound healing, skin appendage regeneration and vascularization. 80 The study concluded that skin appendages and efficient wound healing is possible based on timing of when the stem cells were cultured.80 Though successful the study stated that more research needs to be done to increase skin appendage regeneration in tissue engineering.
- In relation to the development of biomaterials for skin regeneration, the reviewer recommends referencing the following articles:
- Kim, Byoung Soo, et al. "3D cell printing of in vitro stabilized skin model and in vivo pre-vascularized skin patch using tissue-specific extracellular matrix bioink: A step towards advanced skin tissue engineering." Biomaterials 168.
Thank you for your feedback. We have taken your suggestion into consideration by including the recommended article in our citations. As can be seen below on pages:19-20.
5.3 Experimental 3D bioprinting printing in tissue engineering.
The technique of bioprinting, which is a recent development in the field of tissue engineering, involves the use of a bioink solution to create 3D printed products. This bioink solution is made up of various biomaterials in a hydrogel form, along with cells that are crucial for constructing a cellular microenvironment. The presences of cells within the bioink is vital for the creation and maintenance of the 3D printed product, particularly when it is intended for tissue engineering aplications.51 Despite the potential of 3D printing in tissue engineering, there remain obstacles in using the technique for skin replacement or wound healing. Specifically, the bioink materials used for skin are often uniform in composition and may not fully replicate the complex ECM found in natural skin.51-53 Kim et al. sought to investigate the use of Skin Derived ECM (S-dECM) bioink and analyze how effective the solution would be for skin tissue engineering.52 They were able to successfully print a full thickness 3D human skin model with S-dECM bioink, which showed stability with minimal skin shrinkage. S-dECM bioink was compared to their control bioink made of type 1 collagen. The control showed extensive contraction during in vitro tissue culture.52 In sequence to testing the stability of full thickness 3D human skin, researchers used the bioink to print pre-vascularized skin patches. The skin patches promoted in vivo skin wound healing when endothelial progenitor cells were added to the bionk along with pairing the bioink with adipose derived stem cells.52 Although successful with their experiment, Kim et al. stated that more research is necessary to create a bioink that may be useful in 3D skin bioprinting.
Reviewer 2 Report
Comments and Suggestions for Authors
A manuscript with a promising title of “Update in the use of biomaterials for wound healing”
Language wise good, only a few minor spelling errors (el line 34 “…the skin as…”, lines 90, 92 “…collogen…”
Unfortunately, I find the scientific height of this manuscript too low to be considered as an (important) contribution/addition to the literature and should not be published in J Bioengineering. Foremost due to the lack of (scientific and clinical) depth.
Abstract’s fine even though one may discuss whether “recently” is correct regarding the use of biomaterials…has improved healing.
The wound healing process can be dividend in different phases. In this manuscript 4 are chosen. One can argue that the proliferation phase should be divided into proliferation and reepithelialisation, thus adding a 5th phase. Especially in a manuscript like this since reepithelialisation techniques are discussed.
Again (line 53) one can argue “In recent years….”, we’re pushing close to 50 years since Rhinewald and Green's (not even cited) keratinocyte culture findings
“The aim of this review is to highlight properties…applications…current… methods….and new…” I can not see that the aims are met in this manuscript.
Lines 66-67 says same thing as line 65
Line 68-71 – must biomaterials really mimic the natural environment to be able to support cell growth?
Some different materials are mentioned but no depth as to their pros and cons. Many sweeping arguments but no substance that adds to knowledge.
Line 88 – “Natural biomaterials are chiefly used for open wound and burn dressings. Their resemblance…may..”. If dressing materials are also to be reviewed a whole plethora of different materials and techniques are missing (eg cellulose).
Line 91-92 – “Collagen is most commonly…MAY be due to…” Isn’t this just what one expects to find in a review, WHY collagen may be the most commonly used? No answer is found in the manuscript.
Line 94/95 “…MAY release…”, “…are beneficial…”, “…MAY be used…”
Line 101 – “…require deep understanding…” well, isn’t that what a review should aim at providing?
Line 104 “..potential candidates…”
Lines 117-119 – “Once wounding occurs keratinocytes will migrate to the injured dermis and aid in building new blood vessels…” What ref#6 really writes: “Initially, keratinocytes are called to the injured dermis and then the angiogenesis occurs.” I have a hard time understanding keratinocytes’ aid in building new blood vessels.
“…once cells are harvested, they must be placed in a scaffold to proliferate..” – This is not true!
Lines 124-126 – If the skin is used on large wounds – is it really a skin substitute then?!
Line 155 –157 “…encourage new skin…”? “Similar to ….Integra is mainly used used as a temporary…for partial thickness wounds and burns”. Integra is a dermal regeneration template, mainly supposed to be used on full-thickness wounds in order to ‘regenerate’ the dermis (ie to give the skin elasticity and durability). It is supposed to become integrated with the host and later covered with a skin graft, thus not a temporary solution. I can’t really say I’ve heard of anyone using it on partial thickness wounds.
Line 158-164 – “AlloDerm may mimic cadaver allograft…”?” AlloDerm IS cadaver allograft! Using cadaver skin has shown…when used with AlloDerm?! …can be used as a temporary substitute….sure but why would you. If you want a temporary solution there are numerous better and cheaper ones than using a supposedly permanent one.
CEA/CSS, well there are more to it than this. No mention of sprayed cultured keratinocytes. For which reason is CEA/CSS useful in congenital neavus? Why 30% TSBA?
CSS can be constructed in far more diverse ways (and contain a numerous of things) other than stated in the manuscript.
The donor site is not usually a dermatome. Probably the most common donor sites chosen are ventral thighs (not inner or buttocks).
The deeper the dermis of graft…the better healing of wound. Well, no. Thicker skin grafts provide more dermis and thus a more “complete” skin with more suppleness and durability. Full-thickness or thick split-thickness grafts usually are more difficult to get graft success but quality is better. However, the donor site becomes more of a problem the thicker the graft.
Autografts are useful also for minor wounds.
My experience is not that xenografts are absorbed (how could they?) when wound heals. More often it detaches and can be cut away, or sometimes/products dissolve with time.
Lines 215-217 – What is really said here, does this statement add any information?
Meshing skin grafts is first and foremost to gain drainage, the expansion is a secondary bonus. The expansion of STSG is near negligible unless 3:1, 6:1 when the grafts become very difficult to handle and the graft take lower. No mentioning in the manuscript of the disadvantages of widely meshed grafts both regarding take rate and aesthetics. Not a word on MEEK-grafts
Not sure I understand what DNA-dependent PK catalytic subunit is or how it is supposed to work in the clinic.
5.2 Poinas did a study on one patient – this may be relevant in a subsection not as its own heading.
Bringing in adhesive hydrofiber in the manuscript is nice, but then numerous other biomaterial dressings should be added.
Vacuum compression dressing – mixing Simaos ‘homemade’ NPWT with proper NPWT is not fair and shows lack of depth.
Not sure what Table 1 serves for kind of purpose. Just some examples of ongoing research – no depth.
GF may be a solution, yes but how? When? How to do it?
Lines 302-310 – Yes analgesia may be required but that’s hardly the first problem?
Lines 326-327 “More research is needed….” This is true but I can’t really see how this review adds to this
Eventhough, Ouriginal detects only about 5% plagiarism VERY same wordings from references can be found in numerous places, eg:
Manuscript
“The epidermis is generally composed of several layers according to keratinocyte morphology.6 Keratinocytes have been studied as a vital cell source for wound healing and investigated into conjunction with biomaterials. Keratinocytes are formed by division germ cells at the basal cell layer (lower skin layer) while migrating to the upper epidermal layer to form a dead cell on the surface of the skin.6”
Ref #6:
“The epidermis is generally composed of several layers according to keratinocytes morphology. In fact, keratinocytes are formed by division germ cells (basal cells) at the basal cell layer while migrating through a granular layer to the upper epidermal layers to form a dead cell on the surface of the skin [13]. ….”
Comments on the Quality of English LanguageSee above
Author Response
Date: 5/7/2023
Re: Update in the Use of Biomaterials for Wound Healing
Manuscript Number: bioengineering-2358887
To: The Editors at Bioengineering:
We thank the reviewers for their careful consideration of our manuscript. Their thoughtful and constructive comments have led to a significantly revised and improved manuscript. Each reviewer’s comments are addressed below, including references to the corresponding changes to the manuscript.
Reviewer comments are provided below in bold black font. Our responses are provided in blue font. New text is in red font.
Thank you for the consideration of this manuscript.
Sincerely,
Dr Michelle Griffin MBChb PhD
Reviewer #2
- Abstract’s fine even though one may discuss whether “recently” is correct regarding the use of biomaterials…has improved healing.
Thank you for reviewing our paper, we removed the word recently from the abstract.
- Again (line 53) one can argue “In recent years….”, we’re pushing close to 50 years since Rhinewald and Green's (not even cited) keratinocyte culture findings.
Thank you for the review, we removed in ‘recent years’.
- The aim of this review is to highlight properties…applications…current… methods….and new…” I can not see that the aims are met in this manuscript.
Thank you for your review, more content was added to meet the aims.
- Lines 66-67 says same thing as line 65
Thank you for your review, as advised Line 65 was deleted form the manuscript.
- Line 68-71 – must biomaterials really mimic the natural environment to be able to support cell growth?
We agree with the reviewer, and have removed mimic and changed the sentence, as can be seen below:
biomaterials must be able to support cell growth and differentiation analogous to the cells’ original microenvironment.
- Some different materials are mentioned but no depth as to their pros and cons. Many sweeping arguments but no substance that adds to knowledge.
Thank you for your comment. The table does include research articles highlighting key concepts of the review article with appropriate citations.
- line 88 – “Natural biomaterials are chiefly used for open wound and burn dressings. Their resemblance…may..”. If dressing materials are also to be reviewed a whole plethora of different materials and techniques are missing (eg cellulose).
Thank you for your feedback. We have taken your suggestion into consideration by including a section on Cellulose in wound healing. As can be seen below on pages:6-7.
2.2 Cellulose in wound healing
Cellulose, which is an insoluble substance responsible for the structure of a plant’s cell wall, is a natural biomaterial that could assist in tissue regeneration. As a polysaccharide made up of numerous chains of glucose, it has garnered attention for its potential in wound healing because it can imitate the tissue properties required for tissue engineering.15 Bacterial cellulose, also known as cellulose produced by bacteria, is a nano-fiber material that is compatible with the human body. Its capability to retain water and maintain its mechanical strength even when wet make it suitable for tissue engineering purposes.16 Cao et al. conducted a study to investigate the effectiveness of using bacterial cellulose in conjunction with human urine derived stem cell media for wound healing in rats. The researchers opted to use human urine-derived stem cells due to the challenges associated with obtaining mesenchymal stem cells.16 The immune system of Sprague Dawley rats was regulated through cyclosporine injections, after which, 2 cm full thickness wounds were inflicted on the dorsum of the rats. The study findings indicated that the combination of bacterial cellulose and human urine derived stem cells yielded better results in terms of wound healing that either material used alone. Furthermore, the application of both materials resulted in an increase of vascularization compared to when they applied individually.16 Despite cellulose being a promising option for wound healing, its solubility can present challenges in wound care .17
- Line 91-92 – “Collagen is most commonly…MAY be due to…” Isn’t this just what one expects to find in a review, WHY collagen may be the most commonly used? No answer is found in the manuscript.
We apologize for the oversight. We have now included a section on collagen scaffolds in wounds healing as can be seen below on page:6.
2.1 Collagen in wound healing
Collagen is a crucial protein involved in skin regeneration. It is a major component of the ECM and plays a significant role in regulating different phases of wound healing.11 When skin is injured, collagen starts to work immediately to initiate the healing process by activating the coagulation cascade.11 It activates platelet aggregation for clot formation and release cytokines to signal cells such as fibroblasts to start the re-epithelialization process, forming a boundary between the wound bed and external environment.11 Collagen used in biomaterials for tissue engineering is primarily derived from animal sources. Research has demonstrated that bovine collagen can improve wound healing when used as a skin substitute. Li et al. conducted an experiment in mice to investigate the wound healing properties of bovine bone collagen oligopeptides.12 The mice were operated on and given either bovine bone collagen or a control vehicle, and the tensile strength after skin regeneration was compared. mRNA expression levels were also analyzed to identify genes related to the wound healing process.12 The experiment revealed that the mice in the treatment group had increased tensile strength and biochemical markers associated with wound healing showing elevated albumin, prealbumin, transferrin and hydroxyproline.12 Additionally, there was a decrease in neutrophil granulocytes in the treatment group, and an increase in blood capillary formation and the appearance of fibroblast.12 While collagen from animal sources has shown potential in aiding wound healing, there is a potential increased risk of infection associated with its use. The use of collagen derived from bovine sources for wound healing carries a risk of prion infection which may be fatal to the recipient.13
Collagen can be harvested from different animal sources such as rat tail tendon, has been studied for its potential to promote wound healing.11 Rat tail tendon is known for its high concentration for type 1 collagen, a protein that plays a crucial role in wound healing process.11,13 The extracted collagen can be utilized to create sponges that adhere to the wound bed and facilitate a suitable environment for skin tissue regeneration.14 These collagen sponges are commonly employed in treatment of ulcers, pressure sores and burns.14 Collagen continues to be a popular substance in the field of biomaterials for tissue engineering due to its low immune activation, high biocompatibility and ease of manipulation.14
- line 94/95 “…MAY release…”, “…are beneficial…”, “…MAY be used…
Thank you for your feedback. We have taken your suggestion into consideration by toning the language of the sentence down as shown below on page:5.
Natural biomaterials are chiefly used for open wound and burn dressings. Their resemblance to ECM may help stimulate healing and upregulate skin regeneration.10 Natural biomaterials include proteins such as collagen, silk, gelatin, and fibrinogen. Collagen is most commonly used of the natural biomaterials.10 This may be due to the vast amount of collagen found in the skin and the rest of the body. Natural biomaterials may also be of animal origin and comprised of animal derivatives such as keratin, bovine serum albumin, and eggshell membrane.10 Animal derivative biomaterials, such as those that incorporate keratin, may release antibiotics and GFs, features that may be beneficial in wound healing applications.10 Cells are another natural source that may be used in tissue engineering.
- Line 101 – “…require deep understanding…” well, isn’t that what a review should aim at providing?
Thank you for your valuable feedback. We have reworded the sentence as can be seen below on page:8.
Furthermore, current skin replacement techniques require understanding of stem cell function once seeded onto biomaterials.7
- Line 104 “..potential candidates…”
Thank you for your valuable feedback we have now deleted potential candidates.
- Lines 117-119 – “Once wounding occurs keratinocytes will migrate to the injured dermis and aid in building new blood vessels…” What ref#6 really writes: “Initially, keratinocytes are called to the injured dermis and then the angiogenesis occurs.” I have a hard time understanding keratinocytes’ aid in building new blood vessels.
Thank you for your review and feedback, we have now expanded this section as can be seen below on page:9.
Keratinocytes in wound healing
The structure of keratinocytes typically determines the various layers that make up outermost layer of the skin.6 These cells have been extensively studied for their role in wound healing and have been investigated in combination with biomaterials. Cells located in the lower skin layer divide to produce keratinocytes, which then shift to the upper layer of skin and become dead cells on the surface of the skin.6 During the healing process keratinocytes play a crucial role in the proliferative stage. They travel to the damaged dermis and release growth factors such as vascular endothelial growth factor (VEGF), transforming growth factor beta, and fibroblast growth factor. These growth factors promote angiogenesis and the production of connective tissue matrix facilitating tissue regeneration.6,21 Collecting keratinocytes and placing them into a scaffold helps to promote their proliferation. A scaffold is a structure, either artificial or natural, that is composed of various biomaterials and supports tissue development by providing a framework for cell attachment and growth.22 It acts as a temporary ECM that mimic structural and functional properties of the surrounding tissue and serves as a guide for cellular processes such as proliferation, differentiation, and tissue remodeling. In essence, a scaffold is a critical component of tissue engineering that provides a three dimensional environment for cells to interact with each other and their surroundings, enabling the formation of functional tissues that can be used to repaired or replaces damaged or diseased tissues.7,22 The cell seeded scaffold can then be placed over the wound to increase wound healing.7 (Figure 2)
- …once cells are harvested, they must be placed in a scaffold to proliferate..” – This is not true!
Thank you for your valuable feedback, this sentence was deleted as advised.
- Lines 124-126 – If the skin is used on large wounds – is it really a skin substitute then?!
Sorry for the confusion we have now removed the sentence below.
The skin that keratinocytes are harvested from, may also be used as skin substitutes on large wounds.
- Line 155 –157 “…encourage new skin…”? “Similar to ….Integra is mainly used used as a temporary…for partial thickness wounds and burns”. Integra is a dermal regeneration template, mainly supposed to be used on full-thickness wounds in order to ‘regenerate’ the dermis (ie to give the skin elasticity and durability). It is supposed to become integrated with the host and later covered with a skin graft, thus not a temporary solution. I can’t really say I’ve heard of anyone using it on partial thickness wounds.
Thank you for your review and feedback, we have now expanded this section as can be seen below and on page:13.
Integra is a cell-free biomaterial consisting of a collagen/glycosaminoglycan matrix with a silicone membrane. The silicone membrane acts as an epidermis while the collagen matrix provides a suitable environment for skin cells to grow. It was initially created to assist patients with severe full thickness burn injuries. It proved beneficial for patients who had burns so extensive that performing an allograft was challenging due to limited donor sites.36 Though originally developed for burn wounds, integra is now also being used to promote skin regeneration for skin trauma. Patients with extensive skin trauma and limited donor sites may benefit from this treatment. The integra may act as a scaffold for a new layer of skin to form on the wound bed, with blood vessels and skin cells migrating to the collagen matrix. The silicone layer of integra increases wound healing by preventing the loss of fluid, helping the wound to close.36,37
- Line 158-164 – “AlloDerm may mimic cadaver allograft…”?” AlloDerm IS cadaver allograft! Using cadaver skin has shown…when used with AlloDerm?! …can be used as a temporary substitute….sure but why would you. If you want a temporary solution there are numerous better and cheaper ones than using a supposedly permanent one.
Thank you for your review and feedback, we have expanded this section as can be seen below on page:13.
4.3 AlloDerm in wound healing
AlloDerm is a type of human acellular dermal matrix that is derived from cadaver human dermis preserved by freeze drying.36 The removal of cells and cellular debris from Alloderm reduces the likelihood of an immune response that may occur with skin transplantation. As a result, Alloderm has a low rejection rate when used.36 The decellularized ECM of Alloderm retains its basement membrane, which acts as a scaffold containing collagen and elastic fibers. When applied to the wound bed, the scaffold allows cells form the wound to migrate and adhere promoting the growth of new tissue.36 Rennekampff et al. investigated the effects acellular human dermis on wound healing in mice. Human keratinocytes were cultured onto a synthetic water-resistant dressing and applied to the full skin defects on athymic mice. The dressing was applied with and without Alloderm.41 Mice in the acellular dermal matrix group showed reduced wound contraction.41 Like Integra, AlloDerm can be used as a skin substitute for wound care and burns.36
- CEA/CSS, well there are more to it than this. No mention of sprayed cultured keratinocytes. For which reason is CEA/CSS useful in congenital neavus? Why 30% TSBA?
Thank you for your review and feedback, we have expanded this section as can be seen below on page:14.
4.2 Cultured Epidermal Autografts
There are two kinds of autologous skin substitutes: cultured epidermal autografts (CEA) and cultured skin substitutes (CSS).36 CEA consist of culturing keratinocytes from a small sample of the patient’s own skin biopsy and transplanting the epidermal sheet to the wound area.42 The epidermal sheet is thought to be useful in providing permanent wound closure in patients with a burn greater than 30% burn surface area.43 CEAs are often preferred over split thickness grafts for larger burns as they can cover a larger surface area and not limited by availability of donor skin. They can be constructed from smaller sample of skin, making them a more viable smaller sample of skin, making them more viable option for extensive burns.43 It is also believed to be useful in patients with a congenital nevus.42 Isolating the keratinocytes on a small sample of the patients skin can be cultured to increase the number of cells . The keratinocytes can be grated on the area of the nevus to promote growth of healthy new skin.44 According to the investigation by Fujito et al., when the procedure was performed after 6 months of age, complete epithelialization was observed. The study concluded that nevus cells remained in the deepest layer of the epidermis.44 Though useful, CEA does come with significant disadvantages. Vig et al. described how keratinocytes may be difficult to culture thus making delivery of keratinocytes to the wound area difficult.36
- CSS can be constructed in far more diverse ways (and contain a numerous of things) other than stated in the manuscript.
Thank you for your review and feedback, we have expanded this section as can be seen on page:15.
Cultured Skin Substitutes (CSS) consist of autologous keratinocytes, fibroblasts, and stem cells cultured from the patient’s own skin biopsy combined with degradable bovine collagen matrix.39 Living skin cells are grown in a laboratory settling before being transplanted onto the patients wound bed.39 The bovine collagen matrix may act as a scaffold mimicking the extracellular matrix of skin.12,39As the cells grow and differentiate within the scaffold, the scaffold can help to guide the cells to create the structure needed whole providing mechanical support.12,39 The cells cultured can also secrete growth factors and cytokines to stimulate tissue growth repair.6 CSS is useful for permanent wound closure in patients with chronic wounds, congenital nevus, and burns with more than 50% burn surface area, chros.39 Similar to CEAs, CSS can be cultured in large quantities using small sample of the patients own skin which is useful in cases where there is a limited donor site.39 Compared to CEA, a disadvantage of CSS is its lengthier preparation time and higher cost.33
- The donor site is not usually a dermatome. Probably the most common donor sites chosen are ventral thighs (not inner or buttocks).
We apologize for the incorrect terminology and have rephrased this section as can be seen below on page:16.
The donor site is usually from another area that the body, such as the ventral thigh, buttocks, or upper arm.
- The deeper the dermis of graft…the better healing of wound. Well, no. Thicker skin grafts provide more dermis and thus a more “complete” skin with more suppleness and durability. Full-thickness or thick split-thickness grafts usually are more difficult to get graft success but quality is better. However, the donor site becomes more of a problem the thicker the graft.
Thank you for your valuable feedback. We have revised the sentence as can be seen below on page:16.
Slow wound healing may be attributed to deficiency or inadequate presence of keratinocytes.36 Keratinocytes have the ability to mature and differentiate while migrating towards the outer surface of the skin. The keratinized layer provides the skin with essential barrier properties.36 Autograft skin grafting contain many keratinocytes and may be useful for wounds lacking enough keratinocytes to heal.36 When an autograft is necessary, a thin layer of skin including the full epidermis and dermis is shaved from a donor site.36
- Autografts are useful also for minor wounds.
Thank you for your valuable feedback. The section was revised as can be seen below on page:16.
Slow wound healing may be attributed to deficiency or inadequate presence of keratinocytes.36 Keratinocytes have the ability to mature and differentiate while migrating towards the outer surface of the skin. The keratinized layer provides the skin with essential barrier properties.36 Autograft skin grafting contain many keratinocytes and may be useful for wounds lacking enough keratinocytes to heal.36 When an autograft is necessary, a thin layer of skin including the full epidermis and dermis is shaved from a donor site.36
- My experience is not that xenografts are absorbed (how could they?) when wound heals. More often it detaches and can be cut away, or sometimes/products dissolve with time.
Thank you for your review and feedback. We have expanded this section as can be seen below on page:16.
A xenograft is a surgical graft from one species to another.33 Xenografts are usually used as temporary grafts for human wounds.33 The graft fuses to collagen from other animals into the wound, which is believed to increase dermal regeneration.33 Animal such as bovine and swine may be used to create xenografts, the earlier animal used in xenografts was a frog.33 An advantage of using a xenograft is that as the immediate availability, since they are derived from an animal source and can be used without waiting for the patient’s own tissue to be harvested.33 This makes xenografts favorable for surgical patients. The downside of xenografts is that it may cause harm or death to the donor animal.42
The skin substitute examples summarized above may be vital for wound healing in patients. So as to further achieve outcomes in wound healing, researchers are experimenting with novel surgical techniques, tissue augmentation, and wound dressing procedures using a diverse set of biomaterials.
- Lines 215-217 – What is really said here, does this statement add any information?
Thank you for your valuable feedback, we have deleted this sentence.
- Meshing skin grafts is first and foremost to gain drainage, the expansion is a secondary bonus. The expansion of STSG is near negligible unless 3:1, 6:1 when the grafts become very difficult to handle and the graft take lower. No mentioning in the manuscript of the disadvantages of widely meshed grafts both regarding take rate and aesthetics. Not a word on MEEK-grafts
Thank you for your review and feedback, we have expanded this section as can be seen below on page:17.
A fairly new surgical technique is the implementation of a two skin layer pattern graft called auxetic skin graft patterns onto the harvested skin graft.43 Split-thickness skin grafts were most commonly used before the application of auxetic skin pattern grafts. Split thickness grafts are typically created by patterning a small portion of skin with incisions that allow for creation of a mesh. The holes within the graft allows for fluid and blood to drain from under the graft reducing the risk of complications such as hematoma or seroma formation.44 Additionally meshing can improve elasticity of the graft making it more adaptable to curved surfaces of the body. Although useful, split thickness grafts mesh pattern may be aesthetically displeasing as the mesh pattern may be more noticeable where the skin is thin such as the hands or face. Additionally, meshing the skin graft may carry a higher risk of contractures or scarring due to the greater trauma of the skin, which can result in unevenness in the surface contour. The area of coverage is also lower compared to auxetic skin grafting.43,44 Gupta et al. published in 2022 described the implementation of auxetic patterns applied to skin grafts, which allowed for expansion of the grafts to greater than three times the original covered area.43 Some of the patterns investigated in this work allowed for over 30 times expansion, far more than that achieved by current graft patterns.43 Uniaxial and biaxial strains were applied to these auxetic pattern grafts to investigate their resilience to force.43 While some auxetic patterns were found to be stronger than others, this study outlined an important development in potential for graft coverage.43
- Not sure I understand what DNA-dependent PK catalytic subunit is or how it is supposed to work in the clinic.
Thank you for your valuable feedback, we have added our DNA dependent PK catalytic subunits work as below:
In the context of experimental tissue transplantation DNA dependent protein kinase catalytic subunit (DNA-PKcs) has been found to play a role in rejection of transplanted organs or tissue. Harrison et al. found that inhibiting DNA-PKcs lead to decrease in rejection of allogenic skin grafts. We believe this adds substance to the experimental section our aims.
- Poinas did a study on one patient – this may be relevant in a subsection not as its own heading.
Thank you for your comment, we have added more to this section as can be seen below on page:21.
5.3 Experimental tissue augmentation In wound healing
Experimental approaches for tissue augmentation have been explored to improve the success of skin substitution by providing additional support and structure to the graft. Some examples of such approaches include use of fetal skin cells, platelet rich fibrin matrix (PRFM), and sheet-type reprocessed micronized acellular matrix. Poinas et al. explored the use of a collagen matrix seeded with fetal skin cells, reporting that the skin cells may be vital in reducing immune rejection of the skin graft on a patient.48 These cells produce growth factors and cytokines which were hypothesized to increase the rate of healing and promote an immunotolerant state, ideal for graft success.48 Augmentation of a collagen matrix with these cells was also thought to assist with pain reduction in the graft site area.48 While this clinical trial only included a single patient, results were promising and included reduced pain and accelerated wound healing.48
PRFM, a concentrated product of platelets that naturally produce fibrin, has also shown efficacy in several areas of wound healing.49 Reksodipurto et al, used autologous PRFM to augment skin grafts and found that it increased type 1 collagen in augmented skin grafts and improved graft take. 49These approaches have potential to improve the success and outcomes of skin substitution but mor research is needed to determine efficacy and safety. An additional method of tissue augmentation in skin substitution is the development in the use of sheet type reprocessed micronized acellular dermal matrix, a freeze dried and pulverized biomaterial that is reprocessed into a sheet from, which has shown promise to improve outcomes compared to typical ADM.50 Hahn et al investigated the use of this novel biomaterial combined with split thickness skin grafting in 41 patients and found that the outcomes were promising in terms of skin loss rate (9.1%) and graft take rate (>90%).50
- Bringing in adhesive hydrofiber in the manuscript is nice, but then numerous other biomaterial dressings should be added.
Thank you for your review, we have added more to this section as can be seen below on pages:22-24.
Wound maintenance with Adhesive Hydrofiber Foam dressing
Maintenance of wound dressings may be crucial in skin substitution through tissue engineering. In an effort to increase graft uptake, decrease discomfort, and decrease medical waste associated with tie-over bolus dressing with splint fixation, a 2022 investigation by Yen et al. considered the use of an adhesive hydrofiber foam skin graft fixation alternative.51 This retrospective clinical study included 44 patients and found that hydrofiber foam was applied simply and resulted in a waterproof, low volume, reliable dressing for skin grafting procedures without requiring a splint. This alternative allowed for early mobilization, increased patient comfort, and decreased medical waste.51
5.5 Maintenance with Molded Silicone Dressing
In regions that require skin grafts but are topographically uneven, Gotting et al. published a 2022 article proposing the use of addition-cured silicone as dressing for skin grafts that allowed for individualized molding.52 The silicone was applied in gel form over the graft region and hardens quickly with suture threads incorporated for increased stability.52 Benefits of this proposed dressing included highly-specific shaping, support and protection for the graft, and creation of a moist environment.52 The authors found that this method was efficacious in these respects, especially in the context of complex graft sites such as the nose, ear, or nasolabial fold.
5.6 Wound maintenance with Vacuum Compression dressing
Another form of wound maintenance is the use of vacuum compression dressings also known as vacuum assisted closures (VAC). VAC is a type of wound dressing that uses negative pressure to promote wound healing. The dressing may consist of a foam or gauze that is placed over the wound which is then covered by and adhesive film. A tube is inserted into the dressing and attached to a vacuum pump which removes the air and creates negative pressure inside the dressing. This helps to remove excess fluid and cellular debris from the wound as well as increase blood flow.53,54 The tie- over technique is a commonly used immobilization technique used to fix a skin graft to the wound bed to achieve postoperative graft fastening and is often compared to VAC.53,54 Several prospective studies considering the use of negative pressure closure in skin grafting have also demonstrated positive results.54 A study by Nakamura et al. in 2018 involving 13 patients found that negative pressure closure increased graft survival, graft stabilization, and shortened operative time compared to the tie-over technique.54 In a second study by Cao et al., published in 2022, these results were recapitulated in a group of 86 patients.55
5.7 Wound maintenance with Hydrogel dressings
Hydrogel dressings are commonly used for chronic wounds. They’re easily manipulated and modified due to their composition. Their water-resistant properties make them favorable in moist environments. Therefore, additional materials such as antibiotics, antivirals, and antifungal medication can be added to dressings increase wound healing.56 Peptide based hydrogels have become favorable due to their durability, their potential to self-assemble, and capacity to replicate the ECM. Guan et al. completed a review detailing the ability of peptide hydrogels to self-assemble based on their physical environment. 57 Peptide based hydrogel self-assembly aids in the construction of biomolecules and nanostructures that may increase bio-signals necessary for wound healing. 57
- Vacuum compression dressing – mixing Simaos ‘homemade’ NPWT with proper NPWT is not fair and shows lack of depth.
Thank you for your review, we cited Simaos for definition purposes deleting the comparison as can be seen below on page:23.
5.6 Wound maintenance with Vacuum Compression dressing
Another form of wound maintenance is the use of vacuum compression dressings also known as vacuum assisted closures (VAC). VAC is a type of wound dressing that uses negative pressure to promote wound healing. The dressing may consist of a foam or gauze that is placed over the wound which is then covered by and adhesive film. A tube is inserted into the dressing and attached to a vacuum pump which removes the air and creates negative pressure inside the dressing. This helps to remove excess fluid and cellular debris from the wound as well as increase blood flow.53,54 The tie- over technique is a commonly used immobilization technique used to fix a skin graft to the wound bed to achieve postoperative graft fastening and is often compared to VAC.53,54 Several prospective studies considering the use of negative pressure closure in skin grafting have also demonstrated positive results.54 A study by Nakamura et al. in 2018 involving 13 patients found that negative pressure closure increased graft survival, graft stabilization, and shortened operative time compared to the tie-over technique.54 In a second study by Cao et al., published in 2022, these results were recapitulated in a group of 86 patients.55
- Not sure what Table 1 serves for kind of purpose. Just some examples of ongoing research – no depth.
Thank you for your comment. The table does include research articles highlighting key concepts of the review article with appropriate citations.
- GF may be a solution, yes but how? When? How to do it?
Thank you for your comment. We have added to the limitations section as can be seen below on page:25.
6.1 Limitation in vascularization
Vascularization is important for adequate wound healing.58 Growth factors (GF) increase vascularization through angiogenesis, and have therefore been applied to skin substitutes in tissue engineering.58 Amirsadeghi et al. believed that if they were able to release growth factors at a steady state from the skin substitute, this would increase angiogenesis and therefore vascularization.58 Though GF may be a solution to lack of vascularization in tissue engineering, the speed of angiogenesis may delay wound healing.58 GF may promote migration and proliferation of endothelial cells, which are the cells found in blood vessels. Growth factors such as VEGF can stimulate production of new blood vessels by inducing endothelial cells to migrate to the site of the wound bed where the skin substitute may be placed.58 Systemic illnesses such as diabetes may also play a role in the lack of vascularization necessary for tissue engineering, VEGF activity may be impaired leading to delayed angiogenesis and wound healing.58 Schumann et al. investigated the decrease in vascularization within tissue engineering procedures on diabetic mice.59 The experiment consisted of using a poly-L-co-glycolide scaffolds and implanting stem cells to increase wound healing. Application of the scaffold on to the dorsum of C57BL/6 diabetic mice showed decrease in vascularization and inhibited the growth of a microvascular network.
- Lines 302-310 – Yes analgesia may be required but that’s hardly the first problem?
Thank you for your review, we have added more limitations to this section as can be seen below on page:25.
6.1 Limitation in vascularization
Vascularization is important for adequate wound healing.58 Growth factors (GF) increase vascularization through angiogenesis, and have therefore been applied to skin substitutes in tissue engineering.58 Amirsadeghi et al. believed that if they were able to release growth factors at a steady state from the skin substitute, this would increase angiogenesis and therefore vascularization.58 Though GF may be a solution to lack of vascularization in tissue engineering, the speed of angiogenesis may delay wound healing.58 GF may promote migration and proliferation of endothelial cells, which are the cells found in blood vessels. Growth factors such as VEGF can stimulate production of new blood vessels by inducing endothelial cells to migrate to the site of the wound bed where the skin substitute may be placed.58 Systemic illnesses such as diabetes may also play a role in the lack of vascularization necessary for tissue engineering, VEGF activity may be impaired leading to delayed angiogenesis and wound healing.58 Schumann et al. investigated the decrease in vascularization within tissue engineering procedures on diabetic mice.59 The experiment consisted of using a poly-L-co-glycolide scaffolds and implanting stem cells to increase wound healing. Application of the scaffold on to the dorsum of C57BL/6 diabetic mice showed decrease in vascularization and inhibited the growth of a microvascular network.
6.2 Limited donor sites
Limitations may arise due to limited area of donors’ sites when performing an autologous skin graft. Patients who need skin grafts due to burns may face this problem. Burn victims may have an inadequate amount of healthy or undamaged skin which can be harvest and used as a graft.66 Wojciech et al. described donor sites as additional wounds in burn cases, and the addition wounds may require analgesic therapy, leave scars during the healing process, and may increase susceptibility to infection.66 Burns usually require a skin substitute for wound healing, therefore a biocompatible skin substitute that can replace autologous skin substitution should be a priority.64,66
6.3) Immune rejection
Immune response remains a major issue when using biomaterials in tissue engineering. The process of transferring biomaterials on to a wound can cause an increase in activity of the patient’s immune system. An important action of the immune system is the removal of cellular debris, and this may happen at the sight of the wound. Immune cells such as macrophages may cause extensive tissue damage and improper healing when using grafts or scaffolds.67,68 Scientist are searching for ways to decrease the immune inflammatory response in tissue engineering. Ma et al. investigated the use of a titanium plate in dental implants to regulate the immune activity of macrophages within the implant. Macrophages have a dual purpose in wound healing.69 M1 macrophages are inflammatory and M2 macrophages are anti-inflammatory, previous research has shown that M2 macrophages aid in tissue regeneration. The titanium implant showed an increase in M2 macrophages in dental implants. This may be a useful technique in regulating immune rejection in the tissue engineering.68,69
6.4 Limitations in Skin appendage regeneration
Tissue engineering may be able to increase regeneration in wound healing, but complications still arise when it comes to regeneration of skin appendages. Skin appendages include nails, hair follicles and glands. Each skin appendage has a specific function and without it, skin may not be able to function the way it normally does. This may lead to decrease in the protection the skin usually provides. 70 Regeneration of sebaceous glands and hair follicles is an area of research that may be the next step for tissue engineering. Zhang et al. sought to use tissue engineered skin substitutes from adult scalp dermal progenitor cells and epidermal stem cells along with a type 1 collagen scaffold on adult nude mice with full thickness wounds, in hope to increase wound healing, skin appendage regeneration and vascularization. 71 The study concluded that skin appendages and efficient wound healing is possible based on timing of when the stem cells were cultured. 71Though successful the study stated that more research needs to be done to increase skin appendage regeneration in tissue engineering.
- Lines 326-327 “More research is needed….” This is true but I can’t really see how this review adds to this.
Thank you for your comment. The table does include research articles highlighting key concepts of the review article with appropriate citations.
- Eventhough, Ouriginal detects only about 5% plagiarism VERY same wordings from references can be found in numerous places, eg:
We apologize and have checked the revised manuscript with Grammarly a plagiarism checker, and it shows plagiarism at 1%. The Report is attached to the manuscript.
Reviewer 3 Report
Comments and Suggestions for Authors
"Update in the Use of Biomaterials for Wound Healing"
This work updates biomaterials for wound healing. While this review is extensive and relatively summarizes well, some more revisions are needed to improve the quality as follows:
1. Part 5. Experimental skin graft techniques needs significant update, as there should be many more approaches using cell constructs for skin, etc.
2. Authors seem to fail to update the recent works on gels and scaffolds such as 3D printed, peptides, etc. for wound healing materials, they need to cite below refs and discuss within text:
- Large full-thickness wounded skin regeneration using 3D-printed elastic scaffold with minimal functional unit of skin, 2022
- Smart 3D Printed Hydrogel Skin Wound Bandages: A Review, 2022
- Peptide nanogels as a scaffold for fabricating dermal grafts and 3D vascularized skin models, 2022.
- Self‐Assembling Peptide‐Based Hydrogels for Wound Tissue Repair, 2022.
3. Please explain the merits and demerits of cell sources.
4. Authors need to highlight also the extracellular vesicles and cell secretome for future biomaterials usage for wound healing, referencing below articles:
- Stem cell-derived exosomes for wound healing: current status and promising directions
- Cell secretomes for wound healing and tissue regeneration: Next generation acellular based tissue engineered products
5. Please describe the mechanical properties of biomaterials needed for wound healing.
6. Authors need to comment on diabetic wound healing also, as this is a challenging condition clinically, referencing some of the recent biomaterials approaches for this, as shown below:
- Therapeutic arteriogenesis by factor-decorated fibrin matrices promotes wound healing in diabetic mice, 2022
- Functional hydrogels for diabetic wound management, 2021.
7. Future directions need significant improvement, please explain more on the hurdles, challenges now and advice for future development of biomaterials.
Comments on the Quality of English Languagen/a
Author Response
Date: 5/7/2023
Re: Update in the Use of Biomaterials for Wound Healing
Manuscript Number: bioengineering-2358887
To: The Editors at Bioengineering:
We thank the reviewers for their careful consideration of our manuscript. Their thoughtful and constructive comments have led to a significantly revised and improved manuscript. Each reviewer’s comments are addressed below, including references to the corresponding changes to the manuscript.
Reviewer comments are provided below in bold black font. Our responses are provided in blue font. New text is in red font.
Thank you for the consideration of this manuscript.
Sincerely,
Dr Michelle Griffin MBChb PhD
Reviewer #3:
1.Experimental skin graft techniques needs significant update, as there should be many more approaches using cell constructs for skin, etc.
Thank you for your valuable feedback. We have incorporated self-assembling skin grafts into our experimental technique sections Additionally we discussed the use of 3D printing skin grafts as can be seen below on page: 19-20.
5.0 Experimental skin graft techniques
A fairly new surgical technique is the implementation of a two skin layer pattern graft called auxetic skin graft patterns onto the harvested skin graft.46 Split-thickness skin grafts were most commonly used before the application of auxetic skin pattern grafts. Split thickness grafts are typically created by patterning a small portion of skin with incisions that allow for creation of a mesh. The holes within the graft allows for fluid and blood to drain from under the graft reducing the risk of complications such as hematoma or seroma formation.47 Additionally meshing can improve elasticity of the graft making it more adaptable to curved surfaces of the body. Although useful, split thickness grafts mesh pattern may be aesthetically displeasing as the mesh pattern may be more noticeable where the skin is thin such as the hands or face. Additionally, meshing the skin graft may carry a higher risk of contractures or scarring due to the greater trauma of the skin, which can result in unevenness in the surface contour. The area of coverage is also lower compared to auxetic skin grafting.46,47 Gupta et al. published in 2022 described the implementation of auxetic patterns applied to skin grafts, which allowed for expansion of the grafts to greater than three times the original covered area.46 Some of the patterns investigated in this work allowed for over 30 times expansion, far more than that achieved by current graft patterns.46 Uniaxial and biaxial strains were applied to these auxetic pattern grafts to investigate their resilience to force.46 While some auxetic patterns were found to be stronger than others, this study outlined an important development in potential for graft coverage.46
5.1 Experimental self-assembling skin graft
Patients that suffer from burns may have difficulty with the use autologous skin grafts. This may be due to lack of extra skin due to extensive burns. An experimental technique of using self-assembling skin grafts have shown promise of combating this problem. Self-assembling skin grafts are autologous skin substitutes that can replace the skin and epidermis during invasive procedures.48 Climov et al. designed a self-assembling skin graft from porcine punch biopsies.49 The stimulation of dermal fibroblast in the punch biopsies caused the production of an ECM and addition of epidermal progenitor cells helped to organize an epidermis.49 Their investigation concluded that self-assembling skin grafts can be used as a replacement to autologous skin grafts, but more research is needed as results were based on gross examination and histology.
5.2 Experimental tissue transplantation with DNA-dependent protein kinase catalytic subunit
DNA-dependent Protein Kinase Catalytic Subunit (DNA-PKcs) is a protein involved with repairing damaged DNA. DNA-PKcs is known canonically for sensing double stranded DNA breaks and repairing damage through non-homologous end joining.50 It is also known for important roles in transcriptional regulation of pro-inflammatory cytokines.50 These roles have prompted interest in the protein for utility in clinical therapies, and application of DNA-PKcs for immunomodulation in skin transplantation demonstrated encouraging results.50 Harrison et al. explored DNA-PKcs inhibition of the immune system in murine models of after applying a allogenic skin graft to the dorsum of mice.50 Mice that were treated with the inhibitor were found to have lower production of cytokines and decreased T lymphocyte infiltration into the graft. Inhibiting DNA-PKcs also resulted in decreased B-cell proliferation and migration as well as anti-donor antibody production.50 Ultimately, treatment with the inhibitor was found to extend allogenic skin graft survival by five days in a fully mismatched major histocompatibility complex (MHC) skin graft model.50
5.3 Experimental 3D bioprinting printing in tissue engineering
The technique of bioprinting, which is a recent development in the field of tissue engineering, involves the use of a bioink solution to create 3D printed products. This bioink solution is made up of various biomaterials in a hydrogel form, along with cells that are crucial for constructing a cellular microenvironment. The presences of cells within the bioink is vital for the creation and maintenance of the 3D printed product, particularly when it is intended for tissue engineering aplications.51 Despite the potential of 3D printing in tissue engineering, there remain obstacles in using the technique for skin replacement or wound healing. Specifically, the bioink materials used for skin are often uniform in composition and may not fully replicate the complex ECM found in natural skin.51-53 Kim et al. sought to investigate the use of Skin Derived ECM (S-dECM) bioink and analyze how effective the solution would be for skin tissue engineering.52 They were able to successfully print a full thickness 3D human skin model with S-dECM bioink, which showed stability with minimal skin shrinkage. S-dECM bioink was compared to their control bioink made of type 1 collagen. The control showed extensive contraction during in vitro tissue culture.52 In sequence to testing the stability of full thickness 3D human skin, researchers used the bioink to print pre-vascularized skin patches. The skin patches promoted in vivo skin wound healing when endothelial progenitor cells were added to the bionk along with pairing the bioink with adipose derived stem cells.52 Although successful with their experiment, Kim et al. stated that more research is necessary to create a bioink that may be useful in 3D skin bioprinting.
5.4 Experimental 3D printing with minimal functional unit of skin
An additional example of the rapid expansion of the use of 3D printing in tissue engineering, is the ability to heal full-large thickness wounds with a minimal amount of skin. This technique was established to bypass the risk that may occur with traditional tissue engineering. Risk such as limited skin available due to burns, or infection.54 Peng et al. added a minimal functional unit of skin to a 3D printed polyactide-co-caprolactone (PLCL) scaffold along with collagen gel. Skin used in this experiment originated from the tail of a rat. The 3D printed PLCL contained nutrients necessary for the stability of the rat skin. 3D printed PLCL with collagen without skin was used as a control. Wounds treated with the experimental treatment healed faster and showed the ability to regenerate skin appendages such as skin hair follicles and sebaceous glands. 54
5.5 Maintenance with 3D printed hydrogel skin wounds
3D printing may also aid with the dressings applied to wounds. Current techniques used for wound dressings aid in wound healing but can negatively impact wound healing if dressings aren’t done properly. Contaminated dressings can lead to infection, necrosis, and sepsis that may increase mortality.55 Bandages may also adhere to the wound, therefore reducing skin migration and healing.55 Therefore, Tsegay et al. investigated the application of 3D printed hydrogel wound dressings in the form of auxetic structures.56 Dressings were developed with a digital light processing printer. Hydrogel dressings were equipped sensors that were capable of monitoring pH levels and glucose concentrations of the wound. The experiment concluded that 3D printed hydrogels were cost effective and would be beneficial to use on patients with chronic wounds, such as diabetes.56
5.6 Experimental tissue augmentation in wound healing
Experimental approaches for tissue augmentation have been explored to improve the success of skin substitution by providing additional support and structure to the graft. Some examples of such approaches include use of fetal skin cells, platelet rich fibrin matrix (PRFM), and sheet-type reprocessed micronized acellular matrix. Poinas et al. explored the use of a collagen matrix seeded with fetal skin cells, reporting that the skin cells may be vital in reducing immune rejection of the skin graft on a patient.57 These cells produce GFs and cytokines which were hypothesized to increase the rate of healing and promote an immunotolerant state, ideal for graft success.57 Augmentation of a collagen matrix with these cells was also thought to assist with pain reduction in the graft site area.57 While this clinical trial only included a single patient, results were promising and included reduced pain and accelerated wound healing.57
PRFM, a concentrated product of platelets that naturally produce fibrin, has also shown efficacy in several areas of wound healing.58 Reksodipurto et al, used autologous PRFM to augment skin grafts and found that it increased type 1 collagen in augmented skin grafts and improved graft take. 58These approaches have potential to improve the success and outcomes of skin substitution but mor research is needed to determine efficacy and safety. An additional method of tissue augmentation in skin substitution is the development in the use of sheet type reprocessed micronized acellular dermal matrix, a freeze dried and pulverized biomaterial that is reprocessed into a sheet from, which has shown promise to improve outcomes compared to typical ADM.59 Hahn et al. investigated the use of this novel biomaterial combined with split thickness skin grafting in 41 patients and found that the outcomes were promising in terms of skin loss rate (9.1%) and graft take rate (>90%).59 In addition to experimental techniques mentioned, wound dressings are crucial in facilitating skin regeneration. They aid in creating a supportive environment for cell growth and tissue regeneration, thus promoting the development of functional and viable skin tissue.
- Authors seem to fail to update the recent works on gels and scaffolds such as 3D printed, peptides, etc. for wound healing materials, they need to cite below refs and discuss within text:
- Large full-thickness wounded skin regeneration using 3D-printed elastic scaffold with minimal functional unit of skin, 2022
- Smart 3D Printed Hydrogel Skin Wound Bandages: A Review, 2022
- Peptide nanogels as a scaffold for fabricating dermal grafts and 3D vascularized skin models, 2022.
- Self‐Assembling Peptide‐Based Hydrogels for Wound Tissue Repair, 2022.
Thank you for your comment. We have taken into consideration the articles you mentioned and have included a discussion on them in our work. As can be seen below on page: 20-21.
5.4 Experimental 3D printing with minimal functional unit of skin
An additional example of the rapid expansion of the use of 3D printing in tissue engineering, is the ability to heal full-large thickness wounds with a minimal amount of skin. This technique was established to bypass the risk that may occur with traditional tissue engineering. Risk such as limited skin available due to burns, or infection.54 Peng et al. added a minimal functional unit of skin to a 3D printed polyactide-co-caprolactone (PLCL) scaffold along with collagen gel. Skin used in this experiment originated from the tail of a rat. The 3D printed PLCL contained nutrients necessary for the stability of the rat skin. 3D printed PLCL with collagen without skin was used as a control. Wounds treated with the experimental treatment healed faster and showed the ability to regenerate skin appendages such as skin hair follicles and sebaceous glands.54
- Please explain the merits and demerits of cell sources.
Thank you for your review, we have updated our work to include information on pros and cons of different cell sources. As can be seen below on page: 9
Keratinocytes are the most abundant cells present in the epidermis, and they are vital for wound healing.6,7 Several studies have evaluated the use of all of these cell types as cell sources for biomaterials for skin wound repair. However, limitations do arise with using cells form a patient such as cancer or limitations in harvest size.
- Authors need to highlight also the extracellular vesicles and cell secretome for future biomaterials usage for wound healing, referencing below articles:
- Stem cell-derived exosomes for wound healing: current status and promising directions
- Cell secretomes for wound healing and tissue regeneration: Next generation acellular based tissue engineered products
Thank you for your feedback, we have incorporated the recommended articles into our review to enhance our content. As can be seen below on pages:10 -12.
3.3 ADSCs in wound healing
ADSCs are mesenchymal stem cells that have shown increasing evidence of being a key factor in wound healing. These pluripotent stem cells are found in fatty depots and may have the ability to release GFs that aid in tissue regeneration.29 The release of exosomes from ADSC has shown promise in reducing inflammation and scar tissue in wound healing.30 Exosomes are extracellular vesicles that may be released from mesenchymal stem cells, and contain proteins that may aid in the healing of chronic wounds.30 ADSCs may be collected through surgical procedures such liposuction from the abdomen, back or thighs. Once collected the ADSCs are cultured and may be used to aid in wound healing due to their plasticity.31 The ADSCs may be conditioned into media and have shown to increase activity of human dermal fibroblast and keratinocytes, successfully aiding in wound healing in large animal models.31 Ozpur et al. were able to generate a skin substitute from ADSCs and Keratinocytes, which increased wound healing in Wistar albino rats with a 2 cm diameter full thickness skin defect.32
4.1 Acellular skin substitutes
Acellular substitutes are biomaterials typically derived from human or animal tissues wherein immunogenic cellular components are removed using decellularization techniques.12 Acellular substitutes still retain their ECM which aids in proliferation, adhesion, and differentiation.37 (Figure 4) Fadilah et al. completed a review on the use of secretomes to build acellular skin substitutes that may be used in wound healing.38 Secretomes is the conditioned media that stem cells are grown in. While stem cells are growing in media, they release GFs, cytokines, ECM, and other molecules that are vital in wound healing. The media can then be added to biomaterials for the manufacturing of acellular skin substitutes. Addition of secretomes to acellular skin substitutes aid increase vascularization and proliferation, and has showed to be useful for burn patients.37,38 Wang et al. showed increase vascularization when using decellularized substitutes rather than conventional stents.37 Current acellular skin substitutes consist of nylon mesh or collagen acting as the dermis and may have silicone membrane as the epidermis.36 They are commercially available; examples of commercial names include Integra and AlloDerm.36
- Please describe the mechanical properties of biomaterials needed for wound healing.
Thank you for your review. We have now defined mechanical properties of biomaterials needed for wound healing. As can be seen below on pages:3-4.
Biomaterials can be used to aid in the process of wound healing by providing physical support, promoting cell growth and migration, and facilitation the formation of new tissue.7-9 To effectively support these processes biomaterials used in wound healing should possess specific mechanical properties. Some key mechanical properties required from biomaterials for skin regeneration include tensile strength, flexibility, porosity, and biocompatibility.9 Tensile strength helps the material withstands mechanical stress without losing shape, while flexibility allows it to conform to the wounds shape and allows for movement without discomfort.9 Porosity may promote cell growth, biomaterials may contain multiple microscopic and macroscopic pores, which improve cell attachment and proliferation, increase blood flow and cell ingrowth, respectively.7,8 Biodegradability is essential for preventing foreign material accumulation in the body and promoting the formation of new tissue as the wound site heals. These mechanical properties for wound healing are tailored to the specific wound site, taking into account size, location and severity.9 There are different biomaterials to choose from based on necessity.
- Authors need to comment on diabetic wound healing also, as this is a challenging condition clinically, referencing some of the recent biomaterials approaches for this, as shown below:
- Therapeutic arteriogenesis by factor-decorated fibrin matrices promotes wound healing in diabetic mice, 2022
- Functional hydrogels for diabetic wound management, 2021.
Thank you for your review, we have added diabetic wounds as a challenge in tissue engineering. As can be seen below on page: 24 .
6.1 Limitation in vascularization
Vascularization is important for adequate wound healing.67 GFs increase vascularization through angiogenesis, and have therefore been applied to skin substitutes in tissue engineering.67 Amirsadeghi et al. believed that if they were able to release GFs at a steady state from the skin substitute, this would increase angiogenesis and therefore vascularization.67 Though GF may be a solution to lack of vascularization in tissue engineering, the speed of angiogenesis may delay wound healing.67 GF may promote migration and proliferation of endothelial cells, which are the cells found in blood vessels. GFs such as VEGF can stimulate production of new blood vessels by inducing endothelial cells to migrate to the site of the wound bed where the skin substitute may be placed.67 Systemic illnesses such as diabetes may also play a role in the lack of vascularization necessary for tissue engineering, VEGF activity may be impaired leading to delayed angiogenesis and wound healing.67 Schumann et al. investigated the decrease in vascularization within tissue engineering procedures on diabetic mice.68 The experiment consisted of using a poly-L-co-glycolide scaffolds and implanting stem cells to increase wound healing. Application of the scaffold on to the dorsum of C57BL/6 diabetic mice showed decrease in vascularization and inhibited the growth of a microvascular network.
- Future directions need significant improvement, please explain more on the hurdles, challenges now and advice for future development of biomaterials.
Thank you for your review. We have expanded the section on future directions in tissue engineering in our work. As can be seen below on page:27.
7.0 Future directions
Tissue engineering using biomaterials has become an important technique for treating wounds and burns, and the field is continuously evolving.73 Researchers are investigating ways to decrease contraction that occurs when using skin substitutes to promote wound healing, as this can provide better outcome for wound healing.73 Additionally, optimizing the cellular delivery vehicle to induce more re-epithelization as the primary outcome of wound healing is an area of focus.81 Biomaterials are continually being developed and modified to create scaffolds that are more effective in promoting cell growth and tissue regeneration. Novel materials such as 3D-printed structures, hydrogels, and acellular tissue are being explored for their use in tissue engineering.10,52,82 Tissue specific engineering approaches are also gaining momentum. By focusing on the unique characteristics and requirements of different types of tissues, researchers can create more effective and efficient strategies.10,83 Stem cell therapies are also showing significant promise in tissue engineering. Researchers are exploring ways to harness the regenerative properties of stem cells and use them to repair or replace damaged or diseased tissues.84 Microfluidic technologies are emerging as a promising technique for generating dynamic and precise microenvironments for cell growth and tissue regeneration.85 By meticulous control over fluid flow and interactions between cells and biomaterials, researchers can have the ability to create customized environments to promote development of tissues.85 With exciting work ongoing to continue improving applications of biomaterials in wound healing, tissue engineering holds promise as a key technique for improving wound healing.
Round 2
Reviewer 1 Report
Comments and Suggestions for Authors
As the reviewers' comments have been well addressed in the revised manuscript, I recommend acceptance.
Author Response
Thank you for your kind comments and review of our manuscript.
Reviewer 2 Report
Comments and Suggestions for Authors
A revised manuscript with a promising title of “Update in the use of biomaterials for wound healing”
Language has deteriorated since ver 1, numerous spelling errors and grammatically errors making text difficult to understand periodically.
Unfortunately, I still find the scientific height of this manuscript too low to be considered as an (important) contribution/addition to the literature and should not be published in J Bioengineering in its present form. Foremost due to the lack of (scientific and clinical) depth. There are just too many sweeping arguments/explanations and singular findings. Eg the word “may” occurs 82 times in the text. To me “may” express possibility, ie it may, or may not, be like what’s written. Hence assumption and not scientifically based data?
Furthermore, skin substitutes and autologous grafting is used intermingled, and I have a hard time understanding autologous skin grafts as skin substitutes.
Also natural and composite materials can be either degradable or non-degradable (not just synthetic ones) (line 85)
Line 88 – I do not understand how FDA has improved the use of….FDA regulates matter or has FDA performed studies/development of wound dressings?
Line 98 – how interpret this line? That natural biomaterials are mostly used for burns or for burns mostly natural biomaterials are used? Aren’t there many other tissue engineering applications outside of wound and burns where natural biomaterials are used and isn’t there several non-natural biomaterials used in wounds/burns?
Line 108 – How is collagen a crucial protein involved in skin homeostasis? I understand if you talk of the integrity of the skin but homeostasis is more often (in medical writing) used to describe the condition of optimal functioning for the organism, usually including body temperature, fluid balance, pH etc. Whether collagen is mostly a support structure or actively involved in homeostasis. I also wonder whether collagen (line 111) activates the coagulation cascade. The cascade is triggered by collagen yes, but does collagen really have an active part? Furthermore (line 112) fibroblasts start the re-epithelialization process…well in that sense that fibs are there but isn’t it the keratinocytes that re-epithelialize?
Line 169 – what is full partial thickness grafts?
Line 185-6 – I do not understand “However, limitations do arise with using cells form a patient such as cancer or limitations in harvest size”?
In section 3 “skin replacement techniques” are introduced, I’m not sure that skin substitutes equal skin replacement and I don’t get how autologous skin grafts can fit under either heading.
Lines 213-215 – already mentioned the phases, redundant.
Line 253-254 – “I do not understand “With the regeneration of the hair follicle, there may be an association with the turnover of the whole skin”.
4.1 starts with that acellular substitutes are biomaterials typically derived from human or animal tissues…decellularized….followed by secretomes that may enhance…continuing with writing about Wang who used decellularized and then finish the section with materials of nylon mesh quickly jumping to AlloDerm (decellularized) and Integra (acellular from animals). There’s like not really a logical thread. This goes for most of the manuscripts. Things are mentioned but it is very difficult to get a grip whether you are talking about things that are used/work commonly or a study that someone done on 1 patient.
4.2 How do you define “skin cells” it seems like skin cells in the manuscript refers to basically any and all cells that are present in the skin. Line 284”…matrix…suitable..skin cells to grow.” Well fibroblasts and other dermal cells repopulate the matrix, keratinocytes do not crawl under the silicone in any substantial number. Integra is a dermal regeneration template used to get a better ‘dermis’, you always have to autograft after you removed the silicone. Making it not really a skin substitute since it doesn’t substitute the skin.
Line 285-6 “…beneficial…performing an allograft is challenging due to limited donor sites”? Doesn’t compute. Also, using Integra still necessitates autografting which still is challenging due to limited donor sites. Integra is not to be used for lack of donor sites (line 288).
Line 289-92 “…act as a scaffold for a new layer of skin to form…” what is a new layer of skin? You will get something that looks and functions as a dermis- is that a new layer of skin? “The silicone layer…increases wound healing…preventing loss of fluid…” In fact the silicone membrane is more of a problem due to its occlusiveness making Integra vulnerable to infections. Haven’t extensively read refs 39 & 40 but I highly doubt that there will be data telling us that the silicones occlusiveness helps the wound to close.
Line 296 – “..immune response ….occur…with skin transplantation.” Sweeping, what immune response do you get with autologous skin transplantation? If other skin transplantation, which/what kind is refered to?
AlloDerm also always needs autgrafting or other source of keratinocytes, thus not correct to call it a skin substitute since it doesn’t substitute the skin. More of dermal regeneration template like Integra.
Both decrease scar contracture since dermis is not allowed to completely heal by scarring but instead the scaffolding gives a more dermis-like structure with less contracture.
Line 311 – what is magical about 30%TBSA? Does CEA not provide permanent wound closure in smaller wounds?
Lines 311-2 – CEAs are often preferred over STSG….as they can cover larger area…” No! CEAs are not preferred over autologous STSG! However, in certain situations such as very large burns you may not have enough healthy skin to use for autgrafting. CEAs close wounds but also give a very fragile epidermis since the dermis is often lacking (due to eg full thickness burns)
Line 313-5 I do not understand.
Lines 316-20 “…keratinocytes can be grafted on the area of the naevus…promote growth of healthy skin…complete epitheialisation..” don’t you need to excise the neavus and then transplant the CEAs? Will there be healthy new skin afterwards? Well, you have removed dermis and epidermis and provide epidermis only..
That culturing and deliver keratinocytes may very well be difficult but that doesn’t make up the disadvantages. More disadvantageus is the bad quality of skin resulting epidermis, The often very low take rate of CEAs and the hypothetical problem with skin cancer.
What is magical with 50%TBSA that makes CSSs useful for those patients but not them with less extensive wounds?
4.7 Autograft skin substitute – is autologous skin grafting really to be considered skin substitute?
Line 347 – “Autograft skin grafting (wording?) contain many keratinocytes…” well, they consist of dermis and complete epidermis so, yes, there are many keratinocytes there…preferably one would want to use full-thickness grafts to cover wounds. However, full-thickness grafts leaves full-thickness wounds that themselves need to be closed. In some areas it is possible by primary sutures bot otherwise STSG are needs. This makes the amount of full-thickness grafts limited. As the next best one one uses STSG which brings both epidermis and parts of dermis to wound that is to be covered. The donor site heals on its own usually within 2 weeks, however scars be result.
351 – “…epidermal grafts are key…” well STSG are the work horse in burns and other wounds.
Lines 353-4 yes, harvesting STSG thins out the dermis on the donor site. However, extremely seldom this is a problem since most of dermis is left.
4.8 Lines 363- “…advantage…immediate availability…this makes xenografts favorable…the downside…cause harm or death to the donor animal” Xenografts can be used as temporary skin substitutes/dresssings/wound primers. They can, however, not substitute the patient’s own skin. Xenografts need also be autologous grafted or provided with keratinocytes. The temporary biological (xenograft) dressing may however improve wound healing.
Line 377-9 yes, meshed grafts leaves more or less visible scars in square-patterns. Often highly visible in people with more colored skin. Doesn’t really matter whether the skin where the graft is put is thin or not.
Lines 484-5 – well tie-over dressings are common and quite useful, numerous other techniques are available besides hydrofibers. Tie-over dressings are hardly such big of a problem. A study with 44 patients doesn’t really add any significant data, you can probably easily find 20 other studies using other dressings instead of hydrofiber.
6.2 The proper name for this is Negative Pressure Wound Therapy, NPWT. VAC is a brand name. The NPWT system from KCI. Should not be used for a technique. Several would disagree with the wording of vacuum compression.
Using NPWT is surely helpful especially in difficult areas such as eg groin, axillae. However in most areas using NPWT to fixate STSG is really to shoot over the target. The cost of NPWT system compared to common dressings for STSG is highly negative for the NPWT.
Lines 563-6 In a paper of skin substitutes it is important to discuss immune rejection. This section could be greatly expanded and more detailed, eg by mentioning shortcomings of many materials in this sense. Why you choose to bring up titanium plates in dental implants here puzzles me.
The manuscript constitutes a good attempt at summarizing biomaterials in wound healing.
However, the manuscript is to sweeping and perfunctory. Important aspects are hardly touched and others like studies with 1 or 12 subjects made a big fuzz about.
Comments on the Quality of English LanguageA revised manuscript with a promising title of “Update in the use of biomaterials for wound healing”
Language has deteriorated since ver 1, numerous spelling errors and grammatically errors making text difficult to understand periodically.
Unfortunately, I still find the scientific height of this manuscript too low to be considered as an (important) contribution/addition to the literature and should not be published in J Bioengineering in its present form. Foremost due to the lack of (scientific and clinical) depth. There are just too many sweeping arguments/explanations and singular findings. Eg the word “may” occurs 82 times in the text. To me “may” express possibility, ie it may, or may not, be like what’s written. Hence assumption and not scientifically based data?
Furthermore, skin substitutes and autologous grafting is used intermingled, and I have a hard time understanding autologous skin grafts as skin substitutes.
Also natural and composite materials can be either degradable or non-degradable (not just synthetic ones) (line 85)
Line 88 – I do not understand how FDA has improved the use of….FDA regulates matter or has FDA performed studies/development of wound dressings?
Line 98 – how interpret this line? That natural biomaterials are mostly used for burns or for burns mostly natural biomaterials are used? Aren’t there many other tissue engineering applications outside of wound and burns where natural biomaterials are used and isn’t there several non-natural biomaterials used in wounds/burns?
Line 108 – How is collagen a crucial protein involved in skin homeostasis? I understand if you talk of the integrity of the skin but homeostasis is more often (in medical writing) used to describe the condition of optimal functioning for the organism, usually including body temperature, fluid balance, pH etc. Whether collagen is mostly a support structure or actively involved in homeostasis. I also wonder whether collagen (line 111) activates the coagulation cascade. The cascade is triggered by collagen yes, but does collagen really have an active part? Furthermore (line 112) fibroblasts start the re-epithelialization process…well in that sense that fibs are there but isn’t it the keratinocytes that re-epithelialize?
Line 169 – what is full partial thickness grafts?
Line 185-6 – I do not understand “However, limitations do arise with using cells form a patient such as cancer or limitations in harvest size”?
In section 3 “skin replacement techniques” are introduced, I’m not sure that skin substitutes equal skin replacement and I don’t get how autologous skin grafts can fit under either heading.
Lines 213-215 – already mentioned the phases, redundant.
Line 253-254 – “I do not understand “With the regeneration of the hair follicle, there may be an association with the turnover of the whole skin”.
4.1 starts with that acellular substitutes are biomaterials typically derived from human or animal tissues…decellularized….followed by secretomes that may enhance…continuing with writing about Wang who used decellularized and then finish the section with materials of nylon mesh quickly jumping to AlloDerm (decellularized) and Integra (acellular from animals). There’s like not really a logical thread. This goes for most of the manuscripts. Things are mentioned but it is very difficult to get a grip whether you are talking about things that are used/work commonly or a study that someone done on 1 patient.
4.2 How do you define “skin cells” it seems like skin cells in the manuscript refers to basically any and all cells that are present in the skin. Line 284”…matrix…suitable..skin cells to grow.” Well fibroblasts and other dermal cells repopulate the matrix, keratinocytes do not crawl under the silicone in any substantial number. Integra is a dermal regeneration template used to get a better ‘dermis’, you always have to autograft after you removed the silicone. Making it not really a skin substitute since it doesn’t substitute the skin.
Line 285-6 “…beneficial…performing an allograft is challenging due to limited donor sites”? Doesn’t compute. Also, using Integra still necessitates autografting which still is challenging due to limited donor sites. Integra is not to be used for lack of donor sites (line 288).
Line 289-92 “…act as a scaffold for a new layer of skin to form…” what is a new layer of skin? You will get something that looks and functions as a dermis- is that a new layer of skin? “The silicone layer…increases wound healing…preventing loss of fluid…” In fact the silicone membrane is more of a problem due to its occlusiveness making Integra vulnerable to infections. Haven’t extensively read refs 39 & 40 but I highly doubt that there will be data telling us that the silicones occlusiveness helps the wound to close.
Line 296 – “..immune response ….occur…with skin transplantation.” Sweeping, what immune response do you get with autologous skin transplantation? If other skin transplantation, which/what kind is refered to?
AlloDerm also always needs autgrafting or other source of keratinocytes, thus not correct to call it a skin substitute since it doesn’t substitute the skin. More of dermal regeneration template like Integra.
Both decrease scar contracture since dermis is not allowed to completely heal by scarring but instead the scaffolding gives a more dermis-like structure with less contracture.
Line 311 – what is magical about 30%TBSA? Does CEA not provide permanent wound closure in smaller wounds?
Lines 311-2 – CEAs are often preferred over STSG….as they can cover larger area…” No! CEAs are not preferred over autologous STSG! However, in certain situations such as very large burns you may not have enough healthy skin to use for autgrafting. CEAs close wounds but also give a very fragile epidermis since the dermis is often lacking (due to eg full thickness burns)
Line 313-5 I do not understand.
Lines 316-20 “…keratinocytes can be grafted on the area of the naevus…promote growth of healthy skin…complete epitheialisation..” don’t you need to excise the neavus and then transplant the CEAs? Will there be healthy new skin afterwards? Well, you have removed dermis and epidermis and provide epidermis only..
That culturing and deliver keratinocytes may very well be difficult but that doesn’t make up the disadvantages. More disadvantageus is the bad quality of skin resulting epidermis, The often very low take rate of CEAs and the hypothetical problem with skin cancer.
What is magical with 50%TBSA that makes CSSs useful for those patients but not them with less extensive wounds?
4.7 Autograft skin substitute – is autologous skin grafting really to be considered skin substitute?
Line 347 – “Autograft skin grafting (wording?) contain many keratinocytes…” well, they consist of dermis and complete epidermis so, yes, there are many keratinocytes there…preferably one would want to use full-thickness grafts to cover wounds. However, full-thickness grafts leaves full-thickness wounds that themselves need to be closed. In some areas it is possible by primary sutures bot otherwise STSG are needs. This makes the amount of full-thickness grafts limited. As the next best one one uses STSG which brings both epidermis and parts of dermis to wound that is to be covered. The donor site heals on its own usually within 2 weeks, however scars be result.
351 – “…epidermal grafts are key…” well STSG are the work horse in burns and other wounds.
Lines 353-4 yes, harvesting STSG thins out the dermis on the donor site. However, extremely seldom this is a problem since most of dermis is left.
4.8 Lines 363- “…advantage…immediate availability…this makes xenografts favorable…the downside…cause harm or death to the donor animal” Xenografts can be used as temporary skin substitutes/dresssings/wound primers. They can, however, not substitute the patient’s own skin. Xenografts need also be autologous grafted or provided with keratinocytes. The temporary biological (xenograft) dressing may however improve wound healing.
Line 377-9 yes, meshed grafts leaves more or less visible scars in square-patterns. Often highly visible in people with more colored skin. Doesn’t really matter whether the skin where the graft is put is thin or not.
Lines 484-5 – well tie-over dressings are common and quite useful, numerous other techniques are available besides hydrofibers. Tie-over dressings are hardly such big of a problem. A study with 44 patients doesn’t really add any significant data, you can probably easily find 20 other studies using other dressings instead of hydrofiber.
6.2 The proper name for this is Negative Pressure Wound Therapy, NPWT. VAC is a brand name. The NPWT system from KCI. Should not be used for a technique. Several would disagree with the wording of vacuum compression.
Using NPWT is surely helpful especially in difficult areas such as eg groin, axillae. However in most areas using NPWT to fixate STSG is really to shoot over the target. The cost of NPWT system compared to common dressings for STSG is highly negative for the NPWT.
Lines 563-6 In a paper of skin substitutes it is important to discuss immune rejection. This section could be greatly expanded and more detailed, eg by mentioning shortcomings of many materials in this sense. Why you choose to bring up titanium plates in dental implants here puzzles me.
The manuscript constitutes a good attempt at summarizing biomaterials in wound healing.
However, the manuscript is to sweeping and perfunctory. Important aspects are hardly touched and others like studies with 1 or 12 subjects made a big fuzz about.
Author Response
Date: May 24th, 2023
Re: Update in the Use of Biomaterials for Wound Healing
Manuscript Number: bioengineering-2358887
To: The Editors at Bioengineering:
We thank the reviewer for their careful consideration of our manuscript. Their thoughtful and constructive comments have led to a significantly revised and improved manuscript. We truly do appreciate their comments a second time and have tried to address all of their comments. We are happy to make any further edits they feel are necessary.
Reviewer comments are provided below in bold black font. Our responses are provided in blue font. New text is in red font.
Thank you for the consideration of this manuscript.
Sincerely,
Dr Michelle Griffin MBChb PhD
Reviewer 2
- A revised manuscript with a promising title of “Update in the use of biomaterials for wound healing” Language has deteriorated since ver 1, numerous spelling errors and grammatically errors making text difficult to understand periodically.
Thank you for your comment. The manuscript underwent editing to enhance its comprehensibility. We have gone through the entire manuscript and reformatted to improve the understanding.
- Unfortunately, I still find the scientific height of this manuscript too low to be considered as an (important) contribution/addition to the literature and should not be published in J Bioengineering in its present form. Foremost due to the lack of (scientific and clinical) depth. There are just too many sweeping arguments/explanations and singular findings. Eg the word “may” occurs 82 times in the text. To me “may” express possibility, ie it may, or may not, be like what’s written. Hence assumption and not scientifically based data?
Thank you for your comment. The manuscript has been improved to include more scientifically based data and removed the word ‘may’ where not necessary or inappropriate.
- Furthermore, skin substitutes and autologous grafting is used intermingled, and I have a hard time understanding autologous skin grafts as skin substitutes.
Thank you for your comment. The manuscript was enhanced to reduce the blending of autologous grafting and skin substitutes. We have moved the sections around to improve the flow of the manuscript.
4.Also natural and composite materials can be either degradable or non-degradable (not just synthetic ones) (line 85)
Thank you for your comment. We have modified the text as required as shown below and on page: 4.
Synthetic and natural biomaterials may be biodegradable or non-biodegradable.
- Line 88 – I do not understand how FDA has improved the use of….FDA regulates matter or has FDA performed studies/development of wound dressings?
Thank you for your comment, we have removed this sentence to avoid confusion.
- Line 98 – how interpret this line? That natural biomaterials are mostly used for burns or for burns mostly natural biomaterials are used? Aren’t there many other tissue engineering applications outside of wound and burns where natural biomaterials are used and isn’t there several non-natural biomaterials used in wounds/burns?
Thank you for your comment. We have removed this sentence to avoid confusion and improved this paragraph to improve the flow of the paragraph.
- Line 108 – How is collagen a crucial protein involved in skin homeostasis? I understand if you talk of the integrity of the skin but homeostasis is more often (in medical writing) used to describe the condition of optimal functioning for the organism, usually including body temperature, fluid balance, pH etc. Whether collagen is mostly a support structure or actively involved in homeostasis. I also wonder whether collagen (line 111) activates the coagulation cascade. The cascade is triggered by collagen yes, but does collagen really have an active part? Furthermore (line 112) fibroblasts start the re-epithelialization process…well in that sense that fibs are there but isn’t it the keratinocytes that re-epithelialize?
Thank you for your comment. We have carefully reviewed your suggestions and have rephrased this paragraph as shown below and on Page: 5.
Collagen is a major component of the ECM and plays a significant role in the different phases of wound healing.11 When skin is injured, the collagen exposure activates the coagulation cascade.11 The collagen fragments mediate neutrophils to initiate immune responses and the inflammatory phase of wound healing.
- Line 169 – what is full partial thickness grafts?
Thank you for your comment, I have rephrased this to the paragraph below and shown on page: 7.
Park et al. conducted a study to investigate the signaling mechanism involved in the use of silk fibroin for wound healing. Silk fibroin scaffolds were observed to upregulate transcription factor NF-KappaB (NF-kB), which in turn increased the expression of genes necessary for wound healing.20
- Line 185-6 – I do not understand “However, limitations do arise with using cells form a patient such as cancer or limitations in harvest size”?
Thank you for your comment, we have removed this sentence.
- In section 3 “skin replacement techniques” are introduced, I’m not sure that skin substitutes equal skin replacement and I don’t get how autologous skin grafts can fit under either heading.
Thank you for your comment, we agree with the reviewer and removed the sentence that skin substitutes equal skin replacement and moved autologous skin grafts to the introduction where it is more appropriate.
- Lines 213-215 – already mentioned the phases, redundant.
Thank you for your comment. We agree with the reviewer and the phrase was removed.
- Line 253-254 – “I do not understand “With the regeneration of the hair follicle, there may be an association with the turnover of the whole skin”.
Thank you for your comment, we have removed this sentence to enhance comprehensibility of the manuscript.
- 4.1 starts with that acellular substitutes are biomaterials typically derived from human or animal tissues…decellularized….followed by secretomes that may enhance…continuing with writing about Wang who used decellularized and then finish the section with materials of nylon mesh quickly jumping to AlloDerm (decellularized) and Integra (acellular from animals). There’s like not really a logical thread. This goes for most of the manuscripts. Things are mentioned but it is very difficult to get a grip whether you are talking about things that are used/work commonly or a study that someone done on 1 patient.
Thank you for your comment. We agree and feel that the single patient is not relevant and removed this section. The paragraph remains an introduction to acellular skin substitutes only and feel this has improved the flow of the manuscript.
- 4.2 How do you define “skin cells” it seems like skin cells in the manuscript refers to basically any and all cells that are present in the skin. Line 284”…matrix…suitable..skin cells to grow.” Well fibroblasts and other dermal cells repopulate the matrix, keratinocytes do not crawl under the silicone in any substantial number. Integra is a dermal regeneration template used to get a better ‘dermis’, you always have to autograft after you removed the silicone. Making it not really a skin substitute since it doesn’t substitute the skin.
Thank you for your comment. We agree we have rephrased this to make it clear only the dermis is added, and an epidermal graft is needed. As seen below and on page: 10.
Once the dermal skin has regenerated a thin epidermal graft is placed. It was initially created for patients with severe full thickness burn injuries.39 Though originally developed for burn surgery, integra is now also being used for skin trauma. The silicone layer of integra allows for wound closure by preventing the loss of fluid.39,40
- Line 285-6 “…beneficial…performing an allograft is challenging due to limited donor sites”? Doesn’t compute. Also, using Integra still necessitates autografting which still is challenging due to limited donor sites. Integra is not to be used for lack of donor sites (line 288).
Thank you for your comment, we have rephrased this to remove the lack of donor sites.
- Line 289-92 “…act as a scaffold for a new layer of skin to form…” what is a new layer of skin? You will get something that looks and functions as a dermis- is that a new layer of skin? “The silicone layer…increases wound healing…preventing loss of fluid…” In fact the silicone membrane is more of a problem due to its occlusiveness making Integra vulnerable to infections. Haven’t extensively read refs 39 & 40 but I highly doubt that there will be data telling us that the silicones occlusiveness helps the wound to close.
Thank you for your comment, reference 39 does support that integra prevents fluid loss.
- Line 296 – “..immune response ….occur…with skin transplantation.” Sweeping, what immune response do you get with autologous skin transplantation? If other skin transplantation, which/what kind is refered to?
Thank you for your comment, we agree with the reviewer and have rephrased the sentence as seen below and on page: 11.
The removal of cells and cellular debris from Alloderm reduces the likelihood of an immune response and has demonstrated low rejection rates.6
- AlloDerm also always needs autgrafting or other source of keratinocytes, thus not correct to call it a skin substitute since it doesn’t substitute the skin. More of dermal regeneration template like Integra.
Thank you for your comment, we agree with the reviewer and rephrased the sentence as can be seen below and on page: 11.
When applied to the wound bed, the scaffold allows the regeneration of a new dermis
- Both decrease scar contracture since dermis is not allowed to completely heal by scarring but instead the scaffolding gives a more dermis-like structure with less contracture.
Thank you for your comment, We have removed reference to scarring with reference to alloderm.
- Line 311 – what is magical about 30%TBSA? Does CEA not provide permanent wound closure in smaller wounds?
Thank you for your comment, we have removed this sentence from the manuscript.
- Lines 311-2 – CEAs are often preferred over STSG….as they can cover larger area…” No! CEAs are not preferred over autologous STSG! However, in certain situations such as very large burns you may not have enough healthy skin to use for autgrafting. CEAs close wounds but also give a very fragile epidermis since the dermis is often lacking (due to eg full thickness burns)
Thank you for your comment, we apologize and have removed this.
- Line 313-5 I do not understand.
Thank you for your comment, we have removed lines 313-315 to increase understanding of the manuscript.
- Lines 316-20 “…keratinocytes can be grafted on the area of the naevus…promote growth of healthy skin…complete epitheialisation..” don’t you need to excise the neavus and then transplant the CEAs? Will there be healthy new skin afterwards? Well, you have removed dermis and epidermis and provide epidermis only.
Thank you for your comment, we have rephrased this as seen below and on page: 11.
CEAs can be constructed from small regions of skin, making them a more viable option for extensive burns.41 CEAs has been shown to be useful in a patient with a congenital nevus.40 CEA was successful use to promote epithelization of the post curettage wound.42 Although useful, culturing keratinocytes for CEA is still difficult and the rate of successful take is still low.6
- That culturing and deliver keratinocytes may very well be difficult but that doesn’t make up the disadvantages. More disadvantageus is the bad quality of skin resulting epidermis, The often very low take rate of CEAs and the hypothetical problem with skin cancer.
Thank you for comment, the sentence was rephrased as can be seen below and on page: 12.
Although useful, culturing keratinocytes for CEA is still difficult and the rate of successful take is still low.6
- What is magical with 50%TBSA that makes CSSs useful for those patients but not them with less extensive wounds?
Thank you for the comment, we have removed this from the manuscript.
- 4.7 Autograft skin substitute – is autologous skin grafting really to be considered skin substitute?
Thank you for your comment, We have moved this section to the introduction.
- Line 347 – “Autograft skin grafting (wording?) contain many keratinocytes…” well, they consist of dermis and complete epidermis so, yes, there are many keratinocytes there…preferably one would want to use full-thickness grafts to cover wounds. However, full-thickness grafts leaves full-thickness wounds that themselves need to be closed. In some areas it is possible by primary sutures bot otherwise STSG are needs. This makes the amount of full-thickness grafts limited. As the next best one one uses STSG which brings both epidermis and parts of dermis to wound that is to be covered. The donor site heals on its own usually within 2 weeks, however scars be result.
Thank you for your comment, we have rephrased this and moved this to the introduction as can be seen below and on page: 3.
When autologous split thickness skin grafts are necessary, a thin layer of skin including the full epidermis and part of the dermis is shaved from a donor site.6 (Figure 2) The donor site is usually from another area that the body, such as the ventral thigh, buttocks, or upper arm.6 Unfortunately, autologous skin grafts are limited by donor site availability. To overcome this, researchers are aiming to tissue engineer skin replacements.
- 351 – “…epidermal grafts are key…” well STSG are the work horse in burns and other wounds.
Thank you for your comment, we agree with the reviewer this is incorrect and was removed.
- Lines 353-4 yes, harvesting STSG thins out the dermis on the donor site. However, extremely seldom this is a problem since most of dermis is left.
Thank you for your comment, we have rephrased this and moved this to the introduction. As can be seen below and on page: 3.
When autologous split thickness skin grafts are necessary, a thin layer of skin including the full epidermis and part of the dermis is shaved from a donor site.6 (Figure 2) The donor site is usually from another area that the body, such as the ventral thigh, buttocks, or upper arm.6 Unfortunately, autologous skin grafts are limited by donor site availability. To overcome this, researchers are aiming to tissue engineer skin replacements.
- 4.8 Lines 363- “…advantage…immediate availability…this makes xenografts favorable…the downside…cause harm or death to the donor animal” Xenografts can be used as temporary skin substitutes/dresssings/wound primers. They can, however, not substitute the patient’s own skin. Xenografts need also be autologous grafted or provided with keratinocytes. The temporary biological (xenograft) dressing may however improve wound healing.
Thank you for your comment, we have removed this section as it was not appropriate for skin substitutes due to a comment earlier. Thank you for pointing this out.
- Line 377-9 yes, meshed grafts leaves more or less visible scars in square-patterns. Often highly visible in people with more colored skin. Doesn’t really matter whether the skin where the graft is put is thin or not.
Thank you for your comment, we agree with the reviewer and have revised the sentence. The sentence can be seen below and on page: 13.
Although useful, meshing may be aesthetically displeasing.
- Lines 484-5 – well tie-over dressings are common and quite useful, numerous other techniques are available besides hydrofibers. Tie-over dressings are hardly such big of a problem. A study with 44 patients doesn’t really add any significant data, you can probably easily find 20 other studies using other dressings instead of hydrofiber.
Thank you for your comment, as per reviewer we have removed this section as hydrofiber are not a big problem.
33.2 The proper name for this is Negative Pressure Wound Therapy, NPWT. VAC is a brand name. The NPWT system from KCI. Should not be used for a technique. Several would disagree with the wording of vacuum compression.
Thank you for your comment, we agree with the reviewer and have used correct term. As can be seen below and on page: 17.
Another form of wound maintenance is Negative Pressure Wound Therapy (NPWT). NPWT is a type of wound dressing that uses negative pressure to promote wound healing. NPWT removes excess fluid and cellular debris from the wound as well as increase blood flow.62,63
- Using NPWT is surely helpful especially in difficult areas such as eg groin, axillae. However in most areas using NPWT to fixate STSG is really to shoot over the target. The cost of NPWT system compared to common dressings for STSG is highly negative for the NPWT.
Thank you for your comment, we agree with the reviewer and have rephrased the paragraph, as seen below and on page : 1.
Another form of wound maintenance is Negative Pressure Wound Therapy (NPWT). NPWT is a type of wound dressing that uses negative pressure to promote wound healing. NPWT removes excess fluid and cellular debris from the wound as well as increase blood flow.62,63 Several prospective studies consider the use of NPWT in skin grafting have demonstrated positive results.63 A study by Nakamura et al., in 2018 involving 13 patients found that negative pressure closure increased graft survival, graft stabilization, and shortened operative time.63 In a second study by Cao et al., published in 2022, these results were recapitulated in a group of 86 patients.64
- Lines 563-6 In a paper of skin substitutes it is important to discuss immune rejection. This section could be greatly expanded and more detailed, eg by mentioning shortcomings of many materials in this sense. Why you choose to bring up titanium plates in dental implants here puzzles me.
Thank you for your comment, we agree with the reviewer and have rephrased the sentence. The rephrased sentence can be seen below and on page: 19.
Ma et al. studied the role of M2 macrophages on titanium dental implants. Titanium implants were found to increase the number of M2 macrophages, leading to modulation of the immune response.77,78
